# Lateral line ablation by ototoxic compounds results in distinct rheotaxis profiles in larval zebrafish

Kyle C. Newton [ID] [1,5✉], Dovi Kacev[2], Simon R. O. Nilsson[3], Allison L. Saettele[1], Sam A. Golden [ID] [3] & Lavinia Sheets [ID] [1,4✉]

The zebrafish lateral line is an established model for hair cell organ damage, yet few studies link mechanistic disruptions to changes in biologically relevant behavior. We used larval zebrafish to determine how damage via ototoxic compounds impact rheotaxis. Larvae were treated with $CuSO_4$ or neomycin to disrupt lateral line function then exposed to water flow stimuli. Their swimming behavior was recorded on video then DeepLabCut and SimBA software were used to track movements and classify rheotaxis behavior, respectively. Lateral line-disrupted fish performed rheotaxis, but they swam greater distances, for shorter durations, and with greater angular variance than controls. Furthermore, spectral decomposition analyses confirmed that lesioned fish exhibited ototoxic compound-specific behavioral profiles with distinct changes in the magnitude, frequency, and cross-correlation between fluctuations in linear and angular movements. Our observations demonstrate that lateral line input is needed for fish to hold their station in flow efficiently and reveals that commonly used lesion methods have unique effects on rheotaxis behavior.

[1] Department of Otolaryngology, Washington University School of Medicine, St. Louis, MO, USA. [2] Scripps Institution of Oceanography, University of California San Diego, La Jolla, CA, USA. [3] Department of Biological Structure, University of Washington, Seattle, WA, USA. [4] Department of Developmental Biology, Washington University School of Medicine, St. Louis, MO, USA. [5] Present address: Department of Fisheries, Wildlife and Conservation Sciences, Coastal Oregon Marine Experiment Station, Oregon State University, Hatfield Marine Science Center, Newport, OR, USA. ✉email: kylecnewton@gmail.com; sheetsl@wustl.edu

The lateral line is a sensory system used by fishes and amphibians to detect water flow. The functional units of the lateral line are neuromasts; bundles of sensory hair cells located externally along the head and body that mechanotransduce low frequency (≤200 Hz) water flow stimuli into electrochemical signals for interpretation by the central nervous system (reviewed in ref. [1]). The lateral line is known to partially mediate rheotaxis, a multimodal behavior[2] that integrates input from visual[3–5], vestibular[6,7], tactile[3,8–10], and lateral line systems[8–10] to facilitate fish orientation and movement of fish with respect to water flow[9,10].

Although the contribution from the lateral line is well established, there is conflicting evidence on whether it is essential for rheotaxis in fish[2,5,9–12]. Inconsistent methodologies used on distinct fish species that differentially rely on the lateral line to mediate swimming behaviors obfuscate the relationship between the lateral line and rheotaxis. In addition, a recent review hypothesized that differences in the spatial characteristics and velocity of the flow stimuli used to assay rheotaxis likely resulted in the disparate results reported in these studies (see Table 1 in ref. [13]).

Zebrafish hair cells demonstrate unique regenerative features that make them an important model for hearing loss research[14–19]. Many studies have focused on mechanistic disruption of hair cell activity, but few have explored the association between disruption and hair cell mediated behavior. Previous researchers have developed several different assays to study rheotaxis in larval fishes[5,20–22]. However, we sought to develop a behavioral assay and analytical methodology that is sensitive, spatiotemporally scalable, and robust enough to be used on a variety of species, ontogenetic stages, sensory modalities, and behavioral responses that are pertinent to biomedical and ecological research.

We therefore investigated how the lateral line contributes to rheotaxis in larval zebrafish and developed a standardized assay that could identify subtle differences in rheotaxis behavior[12]. We compared the rheotaxis response of fish with an intact lateral line to those with lateral line hair cells ablated by two commonly used compounds: copper sulfate ($CuSO_4$)[17,23–25] and neomycin[5,15,26]. We hypothesized that different ototoxic compounds might produce distinct changes in rheotaxis behavior because they injure lateral lines via different cellular mechanisms[27], and that these behavioral changes could be empirically quantified using machine vision and learning technology.

We found that lateral line ablation by neomycin and $CuSO_4$ did not eliminate rheotaxis behavior but affected the behavioral profiles of larval zebrafish in both general and compound-specific ways. While fish with an intact lateral line could efficiently and effectively maintain their position near the source of the water flow, lateral line ablated fish occupied the rear portion of the working section, performed more rheotaxis events of shorter duration and traveled greater distances. Furthermore, exposure to ototoxic compounds reduced the temporal and angular variation in the swimming kinematics such that lateral line ablated larvae performed rheotaxis with more movements of greater intensity but with reduced efficacy. These results support that sensory input from lateral line organs contribute to efficient positive rheotaxis behavior by allowing fish to detect subtle changes in water flow and to respond with greater economy of movement.

## Results and discussion

To determine the contribution of the lateral line to rheotaxis in fishes, we used $CuSO_4$ and neomycin to ablate the lateral line neuromasts of larval zebrafish (6–7 days post-fertilization (dpf)), then video recorded the swimming behavior of individual larvae in a microflume under no-flow and flow stimulus conditions (Fig. 1).

Our procedures eliminated visual and linear acceleration cues (see methods) and young larval zebrafish cannot detect horizontal angular velocity cues (i.e., yaw[28,29]); however, it was not possible to selectively block tactile cues in a non-invasive manner. To confirm ablation of lateral line neuromasts, a subset of all fish that had undergone behavioral testing were fixed, then processed for immunolabeling of hair cells and innervating afferent neurons (Fig. 2). We observed near total hair cell loss in both anterior and posterior lateral line neuromasts with $CuSO_4$ treatment (Fig. 2a, b, d, e, g) and both a significant impact on neuromast morphology and substantial reduction in hair cell number per neuromast following neomycin treatment (Fig. 2a, c, d, f, g). These results support that the ototoxic compound-treated fish used in our behavior assay had a total absence or severe impairment of lateral line function.

We used machine vision and learning software for 3D pose estimation, movement tracking, and annotation of videos for positive rheotaxis events (as defined in methods) when fish oriented toward (0° ± 45°) and actively swam into the oncoming flow (e.g., Supplementary Fig. 1 and Supplementary Movies 1–3). We standardized our analyses by comparing rheotaxis data acquired during flow to the swimming behavior of fish when they were randomly oriented at 0° ± 45° under no flow (see data analysis). Under no flow conditions, we determined that each treatment group of fish was randomly distributed (Fig. 3 and

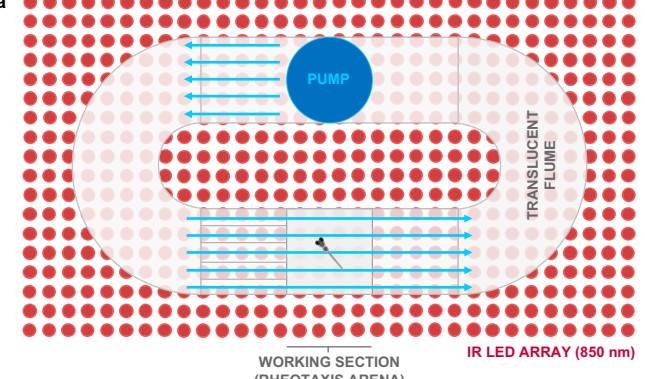

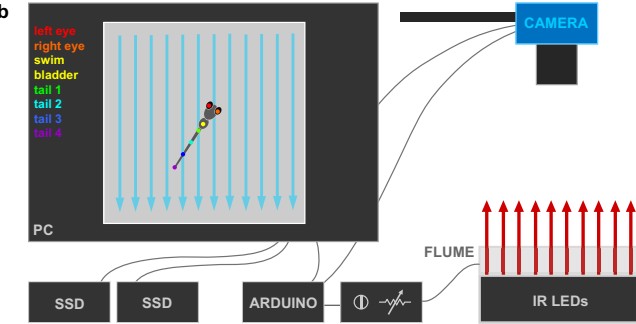

**Fig. 1 Experimental microflume used to conduct rheotaxis assays under IR illumination. a** The microflume (220 × 100 × 40 mm) with a removeable working section (30 × 30 × 10 mm) was 3D printed from translucent resin and placed on top of an infrared (850 nm) LED array. **b** Schematic of experimental set-up. The IR light passed through the flume and the overhead camera recorded rheotaxis trials at either 200 or 60 fps onto SD cards. The timing and duration for the camera and flume pump onset and offset of was controlled by an Arduino and pump voltage (i.e., water flow velocity = 9.74 mm s$^{-1}$) was controlled by a rheostat. Each trial was monitored via the live camera feed displayed on the PC and all videos were copied in duplicate onto a 12TB RAID array. Dots on fish larva indicate seven body positions tracked by DeepLabCut.

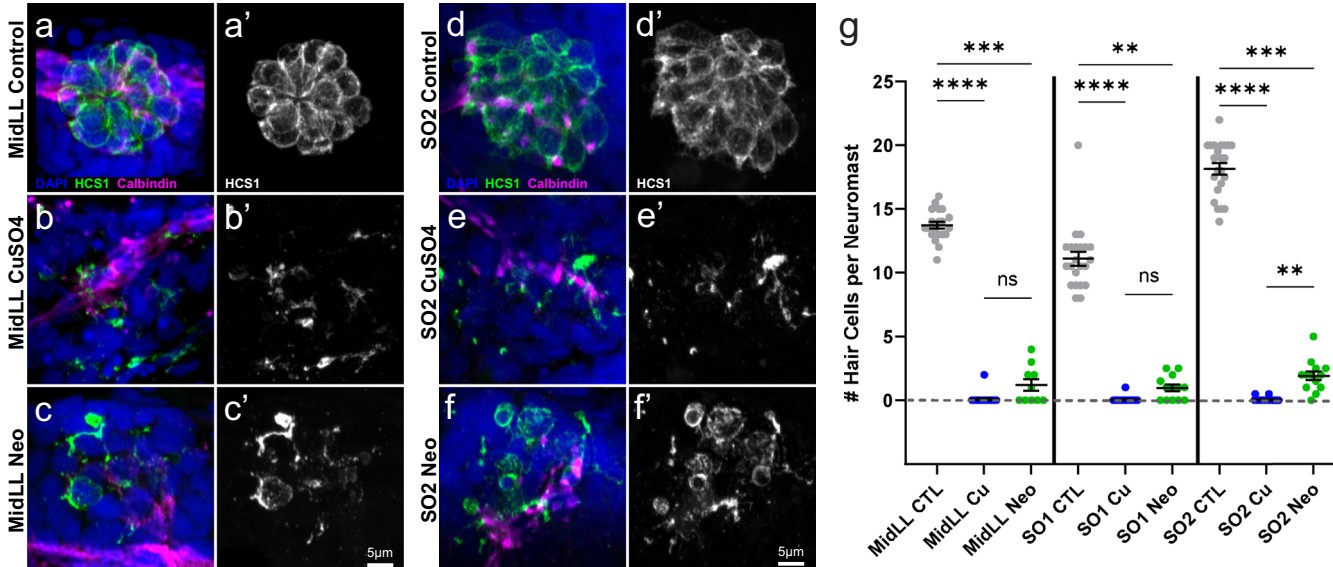

**Fig. 2 Confirmation of neuromast hair cell loss following CuSO$_4$ or neomycin treatment. a–f** Representative confocal max intensity projection images of the: **a–c**) mid posterior lateral line (MidLL) fourth neuromast (L4); and **d–f** second anterior supraorbital (SO2) neuromast from the fish cohorts used for behavior experiments. Hair cells were labeled with an antibody against Otoferlin (HCS1; green **a–f**, gray **a′–f′**). Afferent neurons were labeled with an antibody against Calbindin (magenta), and cell nuclei were labeled with DAPI (blue). **g** Quantification of the grand mean (±SEM) number of hair cells per neuromast in intact (CTL), CuSO$_4$- and neomycin-treated fish. Each dot represents the mean number of hair cells from the MidLL (L3, L4, and L5) or SO (left and right) neuromasts from an individual fish. Data were collected from fish used in three experimental behavior trials; $n = 4$–6 fish per condition per trial. Significance values: **<0.01, ***<0.001, ****<0.0001.

Supplementary Table 1) and had no natural proclivity to orient their bodies to 0° ± 45° (Supplementary Table S2).

**Lateral line ablation alters but does not eliminate rheotaxis behavior**. We predicted that fish treated with the minimum dosage of CuSO$_4$ or neomycin necessary to ablate lateral line hair cells would not perform rheotaxis as well as control fish with an intact lateral line. Surprisingly, fish with lesioned lateral line organs could still orient into oncoming flow like control fish (Fig. 3 and Supplementary Tables 1-2), indicating that the effects of ablation were subtle, and the lateral line was not essential for rheotaxis behavior.

By analyzing the kinematic components of rheotaxis movement, stark behavioral changes were observed among lesioned fish. Comparisons between lateral line-ablated and control groups showed significant differences in the mean body angle or angular variance (Fig. 3 and Supplementary Table 3), mean duration of rheotaxis events, mean number of rheotaxis events, total distance traveled, and latency to the onset of rheotaxis (Fig. 4 and Supplementary Tables 4–7). These observations demonstrate that rheotaxis occurs but is altered in lateral line-ablated groups, which contrasts with previous reports on larval zebrafish[5,22,30] and supports the idea that the lateral line is not required for rheotaxis in fishes[2,11,12,31,32].

We posit that our focus on acute rheotaxis behavior (~20 s) and the fine scale spatiotemporal sampling of machine vision technology allowed us to detect subtle changes in behavior that were overlooked in previous rheotaxis studies[2,5,12,31,32]. Our methods eliminated turbulent water flow, optic flow[3], and certain vestibular cues (yaw: refs. [28,29]; linear acceleration). However, we did not eliminate tactile cues because we could not prevent fish from contacting the substrate and our flow rate was sufficient to displace substrate coupled fish along the bottom and against the rear mesh of arena. Therefore, we propose that lateral line ablated fish might have used tactile cues to gain an external frame of reference and perform rheotaxis[8,10], but explicitly testing this idea

would require designing an experimental apparatus that enables the video capture of fish movements along the vertical plane (Z-axis), which is not possible in our translucent micro flume.

**Ototoxic compounds differentially influence the distribution of mean body angles during flow**. To identify differences in body orientation, the mean body angle for fish in the presence or absence of flow stimuli were compared among treatment groups (Fig. 3). Since flow originated at the top of the flume (located at 0°), mean body angle was determined relative to that of the oncoming water stimulus. The angular variance of each group is represented by the inverse of the mean length of the resultant vector (i.e., short vectors = high variance, and vice versa; Fig. 3). Under no flow, fish from each treatment group swam with a random orientation, indicated by a grand mean body angle that was statistically different from 0° and length of the resultant vector close to zero (Fig. 3a and Supplementary Table 2). When flow was applied, all groups exhibited significant alignment into the oncoming flow stimulus because the grand mean body angles were clustered at 0° and length of the resultant vectors were close to one (Fig. 3b, c).

During rheotaxis, post-hoc comparisons within treatment groups showed a significant difference in the homogeneity of distributions between initial and final portions of the flow stimulus, indicating that overall orientation behavior within groups was not consistent for the duration of flow presentation (Fig. 3b, c and Supplementary Table 3). Comparisons among groups within the initial 10 s portion of flow showed a significant difference in the distribution of mean body angles between control and CuSO$_4$ fish, and between control and neomycin fish (Fig. 3b and Supplementary Table 3). There was an interaction between treatment and stimulus where the distribution of mean body angle in CuSO$_4$-ablated fish during the initial stimulus bin was different than that of neomycin-ablated fish during the final stimulus bin (Supplementary Table 3), suggesting that these lesion methods differentially impacted the rheotaxis behavior of fish.

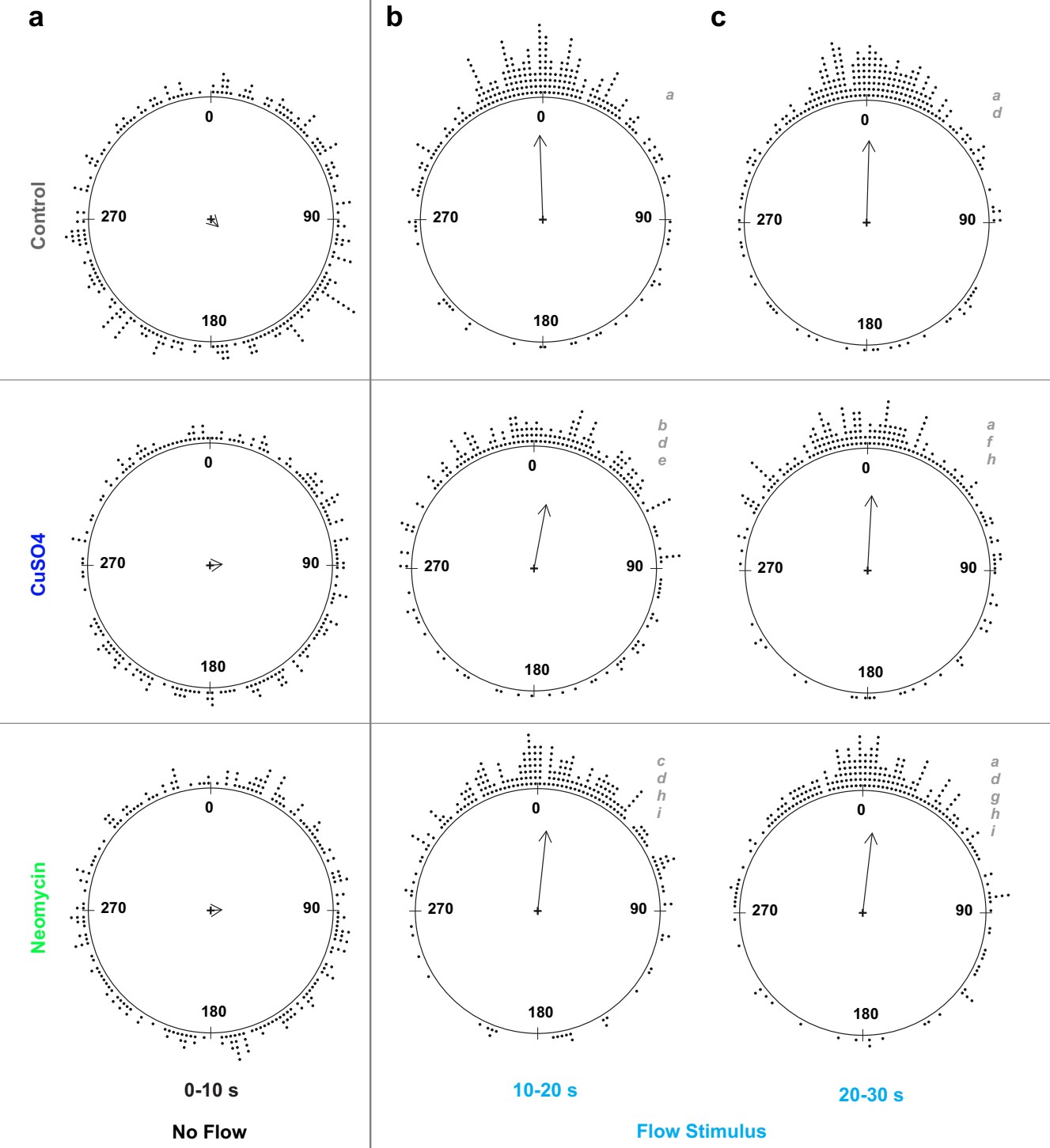

**Fig. 3 The mean resultant vectors of fish treatment groups before and during water flow stimulus indicates fish with lateral-line organs ablated by CuSO₄ or neomycin can still perform rheotaxis. a** Under no flow ($t = 0$-$10$ s), groups of lateral line intact (control, $n = 248$) and lesioned fish (CuSO4, $n = 204$; neomycin, $n = 222$; 18 experimental sessions) have a random distribution of individual mean body angles. **b, c** Under flow, all groups show a statistically significant orientation to $0°± 45°$, but the distributions of the individual mean angles within the groups differ between the initial (**b**; $t = 10$-$20$ s) and final (**c**; $t = 20$-$30$ s) stimulus bins. Each dot outside the circles represents the mean body angle of an individual fish for the 10 s duration of the no flow (**a**) and flow stimulus conditions (**b, c**). The grand mean vector for each group is represented by a summary vector with an angle, *theta*, and a mean resultant length, *rho*, where the length of the vector represents the distribution of individual angles around the mean angle of the group. The length of the vector ranges from zero for uniform distributions, to one for distributions perfectly aligned with the mean angle. Consequently, the angular variance (1-*rho*) is inversely related to vector length. Distributions with the same lowercase letter indicate groups that do not differ statistically.

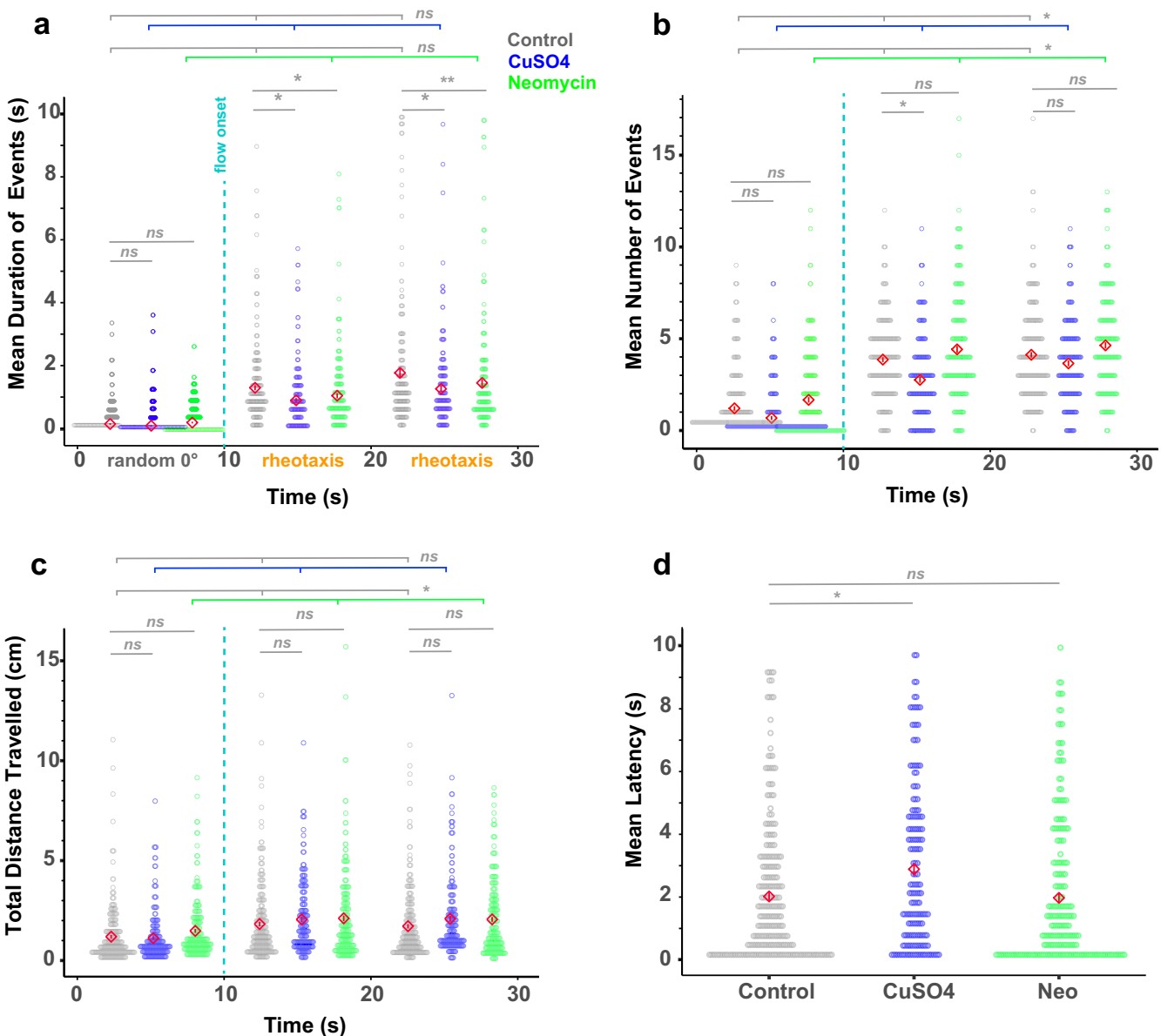

**Fig. 4 Lateral line ablated fish performed rheotaxis for shorter mean durations yet traveled greater total distances compared to controls.** Red diamonds in each plot indicate the mean ± SE values. **a** Lateral line intact fish (gray = control, $n = 248$ fish) have a longer mean duration of rheotaxis events during flow stimulus than lesioned fish (blue = CuSO₄, $n = 204$ fish; green = neomycin, $n = 222$ fish; 18 experimental sessions). **b** The mean number of 0° orientation and rheotaxis events was greatest for neomycin fish and the least for CuSO₄ fish under no flow and flow conditions, respectively. **c** Under no flow, neomycin fish traveled a greater total distance than control and CuSO₄ fish; but under flow, neomycin and CuSO₄ fish traveled a greater total distance than control fish. **d** Compared to control and neomycin fish, CuSO₄ fish had the longest mean latency to the onset of the first rheotaxis event after flow stimulus presentation. Lines indicate statistical comparisons between control and treatment groups (see Supplementary Tables 4–7). The effects of treatment are indicated by long color-coded bars with branches, whereas interactions are indicated with short bars. Significance values: *=0.05, **=0.01.

**Lateral line-ablated fish performed rheotaxis for shorter durations but traveled longer distances.** Although an intact lateral line was not required to perform rheotaxis, lesioned fish behaved differently than non-lesioned fish in flow stimulus, indicating that lack of input from neuromast hair cells affected rheotaxis behavior. To quantify differences in the gross metrics of rheotaxis behavior among groups, we used generalized linear mixed models (GLMMs) that accounted for the random effects of individual variation to compare the mean duration and mean number of rheotaxis events, total distance traveled, and latency between flow presentation and behavior onset. We standardized the data by comparing events of rheotaxis to events when fish

were randomly oriented at 0° ± 45° under no flow. Hereafter both conditions are referred to as *events*.

Without flow, the mean duration of random orientation events did not differ among treatment groups (Fig. 4a and Supplementary Table 4). With flow, the mean duration of rheotaxis events increased for each group, and was greatest in the controls, less in neomycin-, and least in CuSO₄-treated fish (Fig. 4a and Supplementary Table 4). An interaction between stimulus and treatment was seen during the initial 10 s of flow in the CuSO₄ treatment cohort, but during the final 10 s the differences among all treatments became significant (Supplementary Table 4). Additionally, there was a significant effect of stimulus and an

effect of treatment on the mean number of events where the mean was greatest in neomycin-treated, less in controls, and least in CuSO₄-treated fish under no flow and flow conditions (Fig. 4b and Supplementary Table 5).

The total distance traveled during events was influenced by the flow stimulus and type of lesion, where neomycin-treated fish traveled further than control and CuSO₄-treated fish in the presence and absence of flow (Fig. 4c and Supplementary Table 6). Compared to control and neomycin-treated fish, CuSO₄-lesioned fish had a longer latency between flow onset and rheotaxis initiation (Fig. 4d and Supplementary Table 7). Altogether, our data show that an intact lateral line allowed fish to sustain rheotaxis into oncoming flow for longer durations yet travel shorter distances, suggesting that ototoxic compounds reduced the economy of movement of larval zebrafish in response to flow. These results support the idea that the lateral line allows epibenthic fish under flow conditions to hold their station with respect to the substrate[2,33].

Interestingly, CuSO₄ and neomycin ablation affected rheotaxis behavior in different ways. CuSO₄ exposure decreased activity, as evidenced by fewer rheotaxis events with greater latency between flow delivery and behavior initiation. This contrasts with the neomycin-exposed fish that exhibited frequent bursts of rheotaxis and traveled longer distances. While zebrafish larvae have been shown to be relatively resistant to the concentration and exposure time of CuSO₄ used[34], subtle differences in behavior could have been a consequence of non-specific neural toxicity. Alternatively, the distinct effects of CuSO₄ and neomycin treatments on the spatiotemporal nature of rheotaxis might have been due, in part, to their different mechanisms of neuromast ablation. In CuSO₄-treated fish, the hair cells were completely ablated, and the supporting cells and afferent neurons were severely damaged[17]. However, on occasion a few hair cells would remain in neomycin-treated fish (Fig. 2c, f). If residual hair cells in the neomycin group retained some functionality despite severe morphological damage, then it is possible that their sensitivity might have been amplified through efferent modulation[35], or influenced by intact supporting cells (Fig. 2c, f) and recruited from a silent state (e.g., ref. [36]), to compensate for reduced sensory input thus resulting in the observed difference in behavior between lesioned groups.

**Overall spatial use of the arena during rheotaxis differs among treatments**. During flow, we observed that lateral line-lesioned fish were often pushed against the back of the testing arena, whereas intact fish were often swimming at the front of the arena near the flow source. To quantify this observation, density plots of the location of fish in the arena throughout the duration of the experiment were generated (Fig. 5 (2D); Supplementary Fig. 2 (1D)) then compared among treatments. The total density of fish in 2D space was parsed out over five regions of interest (ROI, Fig. 5a) that were determined by the size of fish, their orientation during rheotaxis, and the dimensions of the flow field.

Under no flow, fish that were placed into the flume generally avoided the middle as they explored the boundaries of the arena; however, CuSO₄-treated fish preferred the middle more and explored the boundaries less than the control or neomycin-treated fish (Fig. 5a and Supplementary Fig. 2). Under flow, generalized linear models (Supplementary Table 8) indicate that the spatial use of fish during rheotaxis differed among all three treatment groups. (Fig. 5d). Fish from all treatment groups performed rheotaxis with a pronounced preference for the sides (Fig. 5d) that mimicked the small lateral gradient in the laminar flow field (Supplementary Fig. 3a). The reduced velocity of the laminar flow gradient along the sides was created by a minute boundary layer of null flow adjacent to the wall that gradually

increased (over ~2–5 mm, front to rear) until it became part of the freestream flow field (i.e., blue arrows in Supplementary Fig. 3). The gradient provided a refuge (e.g., ref. [37],) where intact and lesioned fish could swim into the flow with reduced energetic cost[38]. As our definition of rheotaxis (see methods; Supplementary Fig. 1d, e) specifies that only fish that moved their tail and had forward body translation into the flow, fish that were in null water flow (i.e. too close to the wall) were excluded from the rheotaxis dataset. Furthermore, control fish frequently performed rheotaxis while maintaining position in the flow near the upper-left corner of the arena whereas lesioned fish primarily occupied the back-right corner (Fig. 5d and Supplementary Fig. 3b). These observations indicate that an intact lateral line enabled larval zebrafish to better station hold while occupying the areas of strongest flow (Supplementary Fig. 3b) and avoid being swept backwards against the rear mesh. The propensity of lesioned fish to use the rear of the arena while performing rheotaxis in the absence of visual[3] and horizontal vestibular[28,29] cues suggests that these larvae might have used tactile cues to provide the external frame of reference necessary to orient and swim against flow[2,8,10]. Unfortunately, our flume design and camera setup prevented us from exploring this possibility.

**Lateral line ablation reduces the proportion of individual fish that perform rheotaxis**. During flow presentation, the control group had the greatest proportion of individual fish that performed rheotaxis during the experiment (as defined in Supplementary Fig. 1) followed by neomycin-treated then CuSO₄-treated groups (Fig. 6). Intact fish plateaued relatively quickly compared to lesioned fish, but the values for all three groups converged during the final few seconds of flow presentation.

**Spectral decomposition shows that lateral line ablation impacts the overall trend and periodic fluctuation in linear and angular movement**. To uncover the impact that lateral line ablation had on the swimming kinematics of fish, we analyzed how the magnitude of the linear (relative distanced moved, relative velocity, relative acceleration) and angular (mean body angle, mean length of the resultant vector) components of movement changed over time. We observed that all treatment groups swam in the burst-and-glide style that is characteristic of larval zebrafish with intact lateral lines, but with noticeable differences in the quality of their movement. Because the lateral line mediates station holding behavior[2,33,38], we postulated that, under flow, the oscillations in relative linear and angular movements for lateral line-intact fish would be smaller in magnitude than those of lesioned fish. The observed time series data (Fig. 7a, d, g and Supplementary Fig. 4a, d) had a seismic appearance where noise masked the underlying signal. Therefore, we removed the random noise (Supplementary Fig. 4g–k) and decomposed the observed datasets into their fundamental large- and small-scale components: the overall trend in movement magnitude during the entire experiment (Fig. 7b, e, h and Supplementary Fig. 4b, e) and the periodicity, or recurring fluctuations in movement magnitude, that occurred during any given second of the experiment (Fig. 7c, f, i). Because the relative movement (Fig. 7c), velocity and acceleration (Supplementary Fig. 4c, f) data showed similar periodic fluctuations, only relative movement is shown in Fig. 7 for visual clarity.

During rheotaxis, the trend among groups was that CuSO₄ ablation reduced the magnitude of relative movement (Fig. 7b) but not the relative velocity or acceleration of fish (Supplementary Fig. 4c, f) compared to control and neomycin groups. CuSO₄- and neomycin-treated fish had a trend of reduced ability to achieve and maintain orientation within flow relative to controls (Fig. 7e). Lesioned fish had mean resultant vectors (*rho*) of longer length

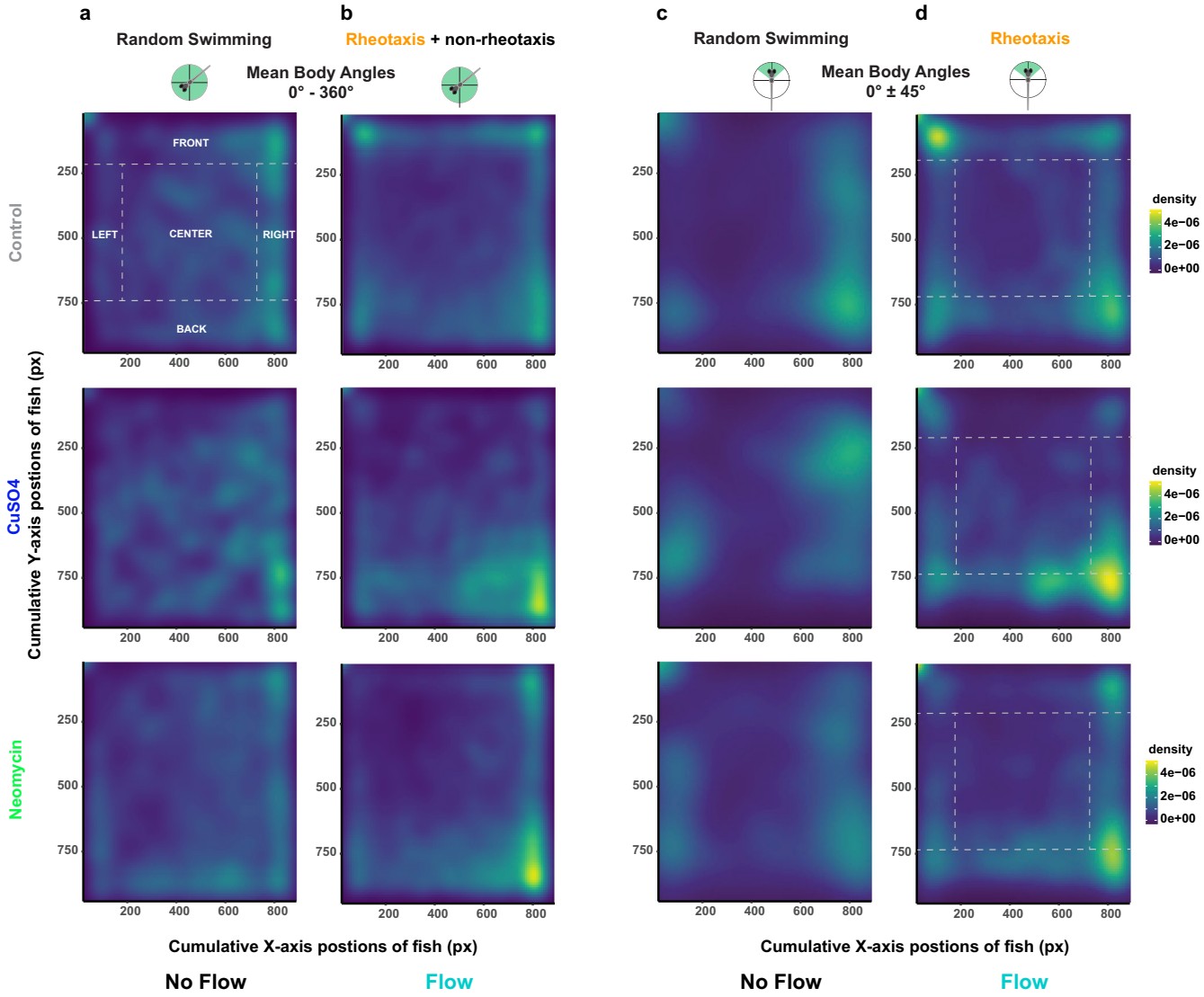

**Fig. 5 Under flow, lateral line intact (control) fish used the front part of the arena near the source of the flow, whereas lesioned (CuSO4, neomycin) fish use the back portion of the arena. a, b** For all possible mean body angles (0°- 360°): **a** all treatment groups of fish show similar density, or total spatial use, of the arena under no flow conditions; **b** however, the controls (control, $n = 248$ fish) used the front of the arena and the lateral-line lesioned fish (CuSO4, $n = 204$ fish; neomycin, $n = 222$ fish; 18 experimental sessions) used the back of the arena under flow. **c, d** Filtering the data for mean body angles required for rheotaxis (0° ± 45°): **c** all groups clustered along the left and right sides of the arena under no flow; **d** but under flow, the controls used the front-left and the lesioned use the back-right portions of the arena. Dotted lines and labels in capital letters indicate spatial regions of interest (ROI). For clarity, statistical comparisons among treatment and ROIs were not included in the figure; however, there were significant fixed effects for CuSO4 and neomycin treatments, the back ROI, and interactions between treatment and ROI (Supplementary Table 8).

meaning that they had less variance (1- *rho*) in their mean body angle compared to controls (Fig. 7h). These data suggest that an intact lateral line allowed control fish to better detect changes in water flow, regularly make small magnitude course corrections to their body angle (Fig. 7h), rapidly orient into flow with greater accuracy (Fig. 7e), which resulted in a greater proportion of non-lesioned fish performing rheotaxis compared to lesioned fish (Fig. 6).

Among treatment groups, there were clear differences in the magnitude of changes in the periodic cycles of linear (Fig. 7c and Supplementary Fig. 4c, f) and angular (Fig. 7f, i) movements during rheotaxis. The linear data show that CuSO4-treated fish had fluctuations of the greatest magnitude in relative distance moved, velocity, and acceleration compared to those of control and neomycin-treated fish (Fig. 7c; Supplementary Fig. 4c, f; and Supplementary Table 9). In CuSO4-treated fish, the amplitude of

the waveform peaks during the sampling period was large and decreased rapidly, but in control and neomycin-treated fish the peaks were small and increased gradually (Fig. 7c and Supplementary Fig. 4c, f). The large fluctuations of CuSO4-treated fish could serve to compensate for their delayed response to flow (Fig. 4d), but when their trend of lesser relative movement (Fig. 7b) is considered, it results in the greatest relative increase in distance traveled among groups (Fig. 4c). These empirical data support the qualitative observations that CuSO4-treated fish performed rheotaxis with delayed responses and erratic linear movements, whereas control and neomycin-ablated fish performed rheotaxis with low latency responses and graded linear movements.

The periodic fluctuation in mean body angle among treatment groups showed an initial small turn to the left (negative values) followed by a series of right (positive values) and left turns that

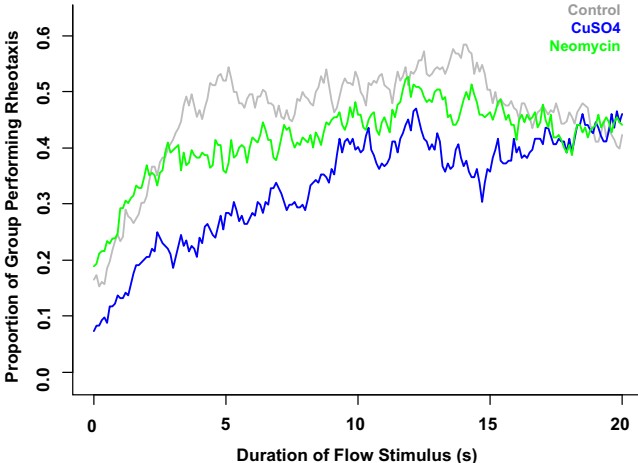

**Fig. 6 Intact lateral line enabled a greater proportion of fish to perform rheotaxis within the first 15 seconds of stimulus onset.** Time series of the proportion of individual fish within each group that performed rheotaxis during flow presentation. A greater proportion of lateral line intact fish (gray = control, $n = 248$ fish) performed rheotaxis than lesioned fish (blue = CuSO$_4$, $n = 204$ fish; green = neomycin, $n = 222$ fish; 18 experimental sessions). The data for all treatments converges after 17 s of flow presentation.

were of relatively small magnitude in controls and CuSO$_4$-treated fish, and relatively large magnitude in neomycin-treated fish (Fig. 7f; Supplementary Table 9). The amplitude of the waveform peaks in control and CuSO$_4$-treated fish were relatively small and gradually increased during the sampling period, but those of neomycin-treated fish were relatively large throughout. Therefore, control and CuSO$_4$-treated fish performed rheotaxis with a graded response in mean body angle, but neomycin-treated fish performed rheotaxis with large erratic changes in mean body angle. We also examined the periodicity in mean resultant length, which reflects the fluctuation in angular variance as fish performed rheotaxis. In control fish, the waveform had peaks of small amplitude that rapidly increased during the sampling period, but in lesioned fish the waveform was comprised of small peaks of consistent amplitude (Fig. 7i). Intact fish showed a graded response where small initial changes in angular variance preceded much larger subsequent changes that coincided with the flat portion of the peaks in mean body angle (Fig. 7f), which indicates that control fish regularly made small changes in their body angle and maintained this new heading before making another angular adjustment. Conversely, lesioned fish had very little fluctuation in angular variance (Fig. 7i) once a mean body angle was chosen, regardless of whether the change in mean body angle was relatively small (e.g., CuSO$_4$) or large (e.g., neomycin; Fig. 7f).

These data support that, during rheotaxis, fish with an intact lateral line organ made small adjustments to their linear movement, velocity, acceleration, and mean body angle with a high degree of variability. This graded, flexible, and finely tuned behavioral response to flow resulted in a greater ability of control fish to rapidly attune and maintain fidelity to the oncoming flow vector (Fig. 7e). Conversely, CuSO$_4$- and neomycin-ablation caused fish to swim into flow with erratic linear and angular movements, respectively, which ultimately undermined their ability to maintain a 0° heading into the flow (Fig. 7e).

**Spectral analysis reveals that lateral line ablation impacts the temporal fluctuation of linear and angular movement.** To determine the impact that lateral line ablation had on the frequency of the fluctuations in linear and angular movement, we

decomposed each periodicity dataset (Fig. 7c, f, i and Supplementary Fig. 4c, f) into a power spectrum that depicted the spectral density (i.e., number of changes in movement per frequency) over a continuous range of frequencies (Fig. 8a–e). For each parameter, the frequency and amplitude of three most dominant peaks were summed to calculate the net shifts in frequency and spectral density among treatment groups (Supplementary Table 10). Relative to controls, the power spectra of CuSO$_4$- and neomycin-treated fish showed a net shift to lower dominant frequencies (downshift) and a net increase in the density of the dominant peaks for relative movement, velocity, and acceleration (Fig. 8a–c and Supplementary Table 10), which indicates that lateral line-ablated fish performed rheotaxis with greater numbers of linear movements to compensate for their overall reduction in fluctuation frequency. The rheotaxis behavior of CuSO$_4$-treated fish showed less frequent fluctuations in relative movement compared to controls and neomycin-treated fish (Fig. 8a), but the fluctuations in relative velocity and acceleration for CuSO$_4$-treated fish and controls occurred over a wider range of frequencies than those of neomycin-treated fish (Fig. 8b, c). Specifically, the relative movement spectra of the control and CuSO$_4$-treated fish had broad clusters of dominant peaks that gradually shifted to higher frequencies (upshifted) for relative velocity and acceleration. However, the spectra of neomycin-treated fish were dominated by a single high-density peak that consistently occurred at 0.21 Hz for relative movement, velocity, and acceleration (Fig. 8a–c). These data reveal that lateral line ablation resulted in rheotaxis behavior characterized by fluctuations in relative movement, velocity and acceleration that occur less frequently, but that neomycin ablation has the additional effect of temporally restricting the frequency of fluctuations in linear movement to occur around a consistent dominant frequency.

Power spectra density curves indicated that intact fish performed rheotaxis with fewer numbers of fluctuations in mean body angle that occurred less frequently, whereas lesioned fish performed rheotaxis with greater numbers of high frequency fluctuations in mean body angle. The spectrum of controls shows five broadly spaced peaks, but those of the lesioned fish have fewer peaks that are clustered around the dominant frequencies (Fig. 8e) indicating that intact fish performed rheotaxis with greater variety in the frequency of fluctuations in their angular variance compared to lesioned fish (Fig. 8e). We observed the mean body angle of CuSO$_4$- and neomycin-treated fish showed a net upshift in the dominant frequencies, a net increase in dominant peak density, and temporal restriction, or greater clustering of power spectra into distinct peaks, relative to that of controls (Fig. 8d and Supplementary Table 10). By contrast, the spectrum of control fish was relatively flat with low density peaks that lacked distinct clustering into dominant frequencies (Fig. 8d) Additionally, there were two divergent patterns seen in the power spectra for mean resultant vector length in lesioned fish: CuSO$_4$-treated fish had a net downshift in dominant frequencies and a net increase in peak density, whereas neomycin-treated fish had a net upshift in frequency and a net decrease in peak density relative to those of controls (Fig. 8e and Supplementary Table 10). Thus, CuSO$_4$-lesioned fish performed rheotaxis with greater number of fluctuations in angular variance that occurred less frequently, but neomycin-lesioned fish performed rheotaxis with fewer numbers of fluctuations in angular variance that occurred more frequently compared to intact fish. These results show that lateral line ablation in larval zebrafish resulted in rheotaxis behavior with greater numbers of fluctuations in mean body angle, fewer numbers of fluctuations in angular variance, and a greater clustering of angular movements into fewer peaks.

We interpret the relatively flat and low-density power spectra observed in control fish (Fig. 8a–d) to indicate that an intact

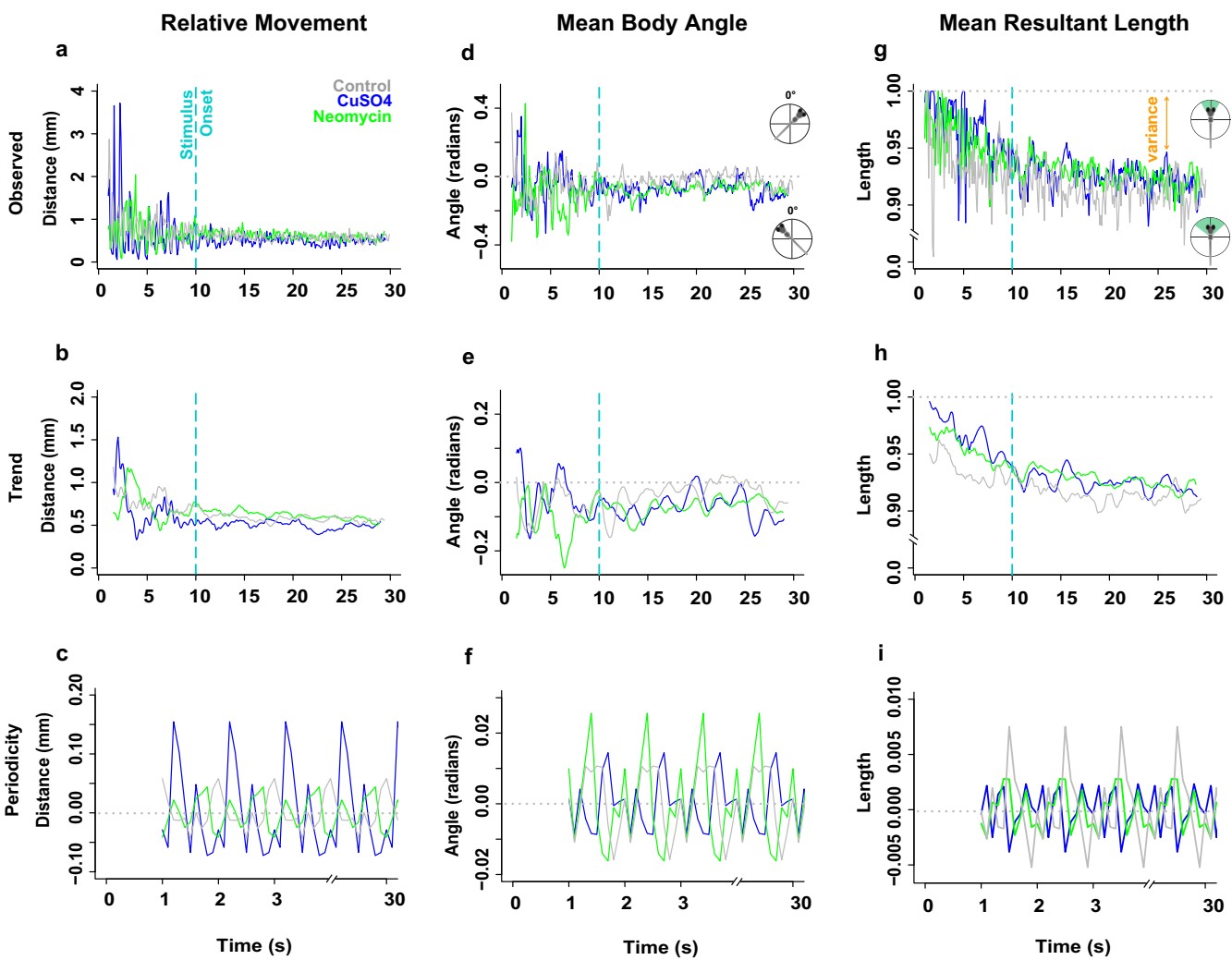

**Fig. 7 The overall trends and periodic fluctuations in the linear (relative distance moved) and angular (mean body angle, mean length of the resultant vector) motion parameters of rheotaxis behavior differ among treatment groups.** (Note: the relative velocity and acceleration periodicity data mimicked the patterns observed in relative movement; see Supplementary Fig. 4). Gray = control ($n = 248$ fish), blue = $CuSO_4$ ($n = 204$ fish), green = neomycin ($n = 222$ fish). Spectral decomposition of the observed data (**a**, **d**, **g**) removed the noise (Supplementary Fig. 3g, j, k) to reveal the overall underlying trends (**b**, **e**, **h**) and the periodicity, or recurring fluctuations (**c**, **f**, **i**) that occurred during any given 1 s of the experiment. The periodicity waveform peaks (**c**, **f**, **i**) indicate the average amount (amplitude), number, direction (positive = increasing; negative = decreasing), and order of occurrence for these cyclic fluctuations as a function of unit time (1 s). The overall trends were that $CuSO_4$ treated fish had the least relative movement (**b**), while the control fish more rapidly oriented to 0° (**e**) and swam with more angular variance (h; 1 − mean length of the resultant vector) compared to lesioned fish. The periodic fluctuation in relative distance moved (**c**) was greatest in $CuSO_4$ treated fish compared to control or neomycin treated fish. However, the fluctuation in mean body angle (**f**) was greatest in neomycin treated fish compared to control and $CuSO_4$ fish, while the fluctuation in mean length of the resultant vector (i.e the angular variance; **i**) was greatest in control fish compared to lesioned fish.

lateral line allowed fish to respond to flow with fewer overall fluctuations in linear and angular movement over a broader range of fluctuation frequencies compared to lesioned fish. It is sensible that the mechanosensory input from lateral line hair cells gave intact fish a greater ability to respond to flow with greater efficiency, efficacy, and economy of movement. Conversely, fewer peaks of greater density observed in lateral line lesioned fish indicates that losing the ability to detect water flow led to more changes in linear or angular movement that occurred over a restricted range of frequencies thus reducing the effectiveness of their movements in response to flow.

**Cross correlation between linear and angular movements shows distinct rheotaxis profiles among groups.** In the absence of visual cues, intact larval zebrafish detected the presence of flow

through a change in linear or angular displacement[22] and responded by either turning right, left, or moving forward into the flow. Because these movements relied on input from the lateral line, we investigated the effect of ototoxic compounds on the cross correlation between linear and angular movements. The correlograms depict the degree, direction, and relative timing between an above average increase in relative movement (Fig. 9a, b), velocity (Fig. 9c, d), or acceleration (Fig. 9e, f) and an above average (i.e. relatively large) change in mean body angle (Fig. 9a, c, e) or mean resultant length (Fig. 9b, d, f). By focusing on the strongest cross-correlations, treatment-specific patterns emerged in the direction and relative timing between large changes in linear and angular movements during rheotaxis.

The rheotaxis movements of control fish in response to flow were smoother, less erratic, and more effective at station holding within the flow field compared to those of lesioned fish (Fig. 9

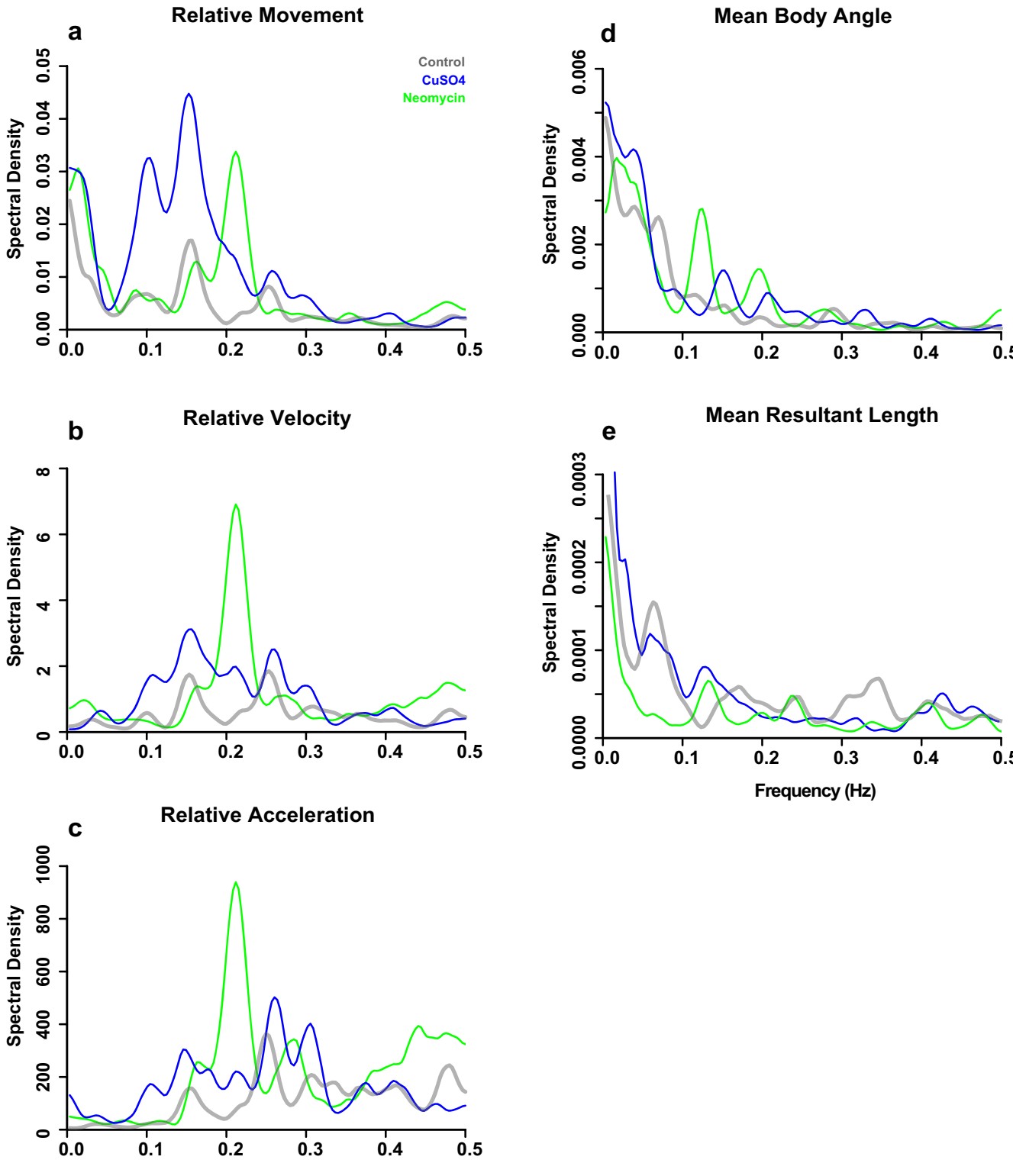

and Supplementary Table 11). The rheotaxis profile of control fish was such that the relative timing and direction of changes in mean body angle depended upon the time derivative of the linear movement, but changes angular variance were consistent across all linear movements. For example, intact fish tended to change their mean body angle to the *left* of flow *before* a large change in relative movement (Fig. 9a), to the *right* of flow *simultaneous* with changes in velocity (Fig. 9c), and to the *left* of flow *after* changes

in acceleration (Fig. 9e). However, the strongest correlation between a *reduction* in angular variance always occurred *prior* to large changes in relative movement (Fig. 9a), velocity (Fig. 9c), and acceleration (Fig. 9e). In lesioned fish, the timing and direction of the strongest cross-correlations shifted the rheotaxis profile of these fish away from that of controls in a predictable, consistent, and treatment-specific manner. CuSO4-treated fish tended to change their mean body angle to the *right* of the

**Fig. 8 Power spectra density curves show that an intact lateral line allowed fish to make fewer yet more temporally variable changes in relative linear and angular movement.** Because frequency and period are inversely related, the low frequency peaks to left of the periodograms indicate cycles with longer periods, and vice versa The amplitude of the peaks indicates the spectral density, or the number of movement events at a given frequency that occurred during the experiment. The peaks with the greatest amplitude indicate the fundamental or dominant frequencies of fluctuation in the periodicity data. The frequency and amplitude of three most dominant peaks were summed to calculate the net shifts in frequency and power. Relative to controls (gray, $n = 248$ fish), lesioned fish (blue = CuSO$_4$, $n = 204$ fish; green = neomycin, $n = 222$ fish) had a net downshift in the three dominant frequencies of **a** relative movement, **b** velocity, and **c** acceleration and a net upshift in **d** mean body angle of larval zebrafish during rheotaxis. For the dominant frequencies of **e** mean resultant length, there was a net downshift and upshift for CuSO$_4$- and neomycin-treated fish, respectively. Furthermore, relative to lateral line intact fish, the peaks of lesioned fish are clustered into fewer peaks of greater amplitude over a relatively narrow range, which indicates that lateral line ablation increased the number yet reduced the temporal variation of changes in movement.

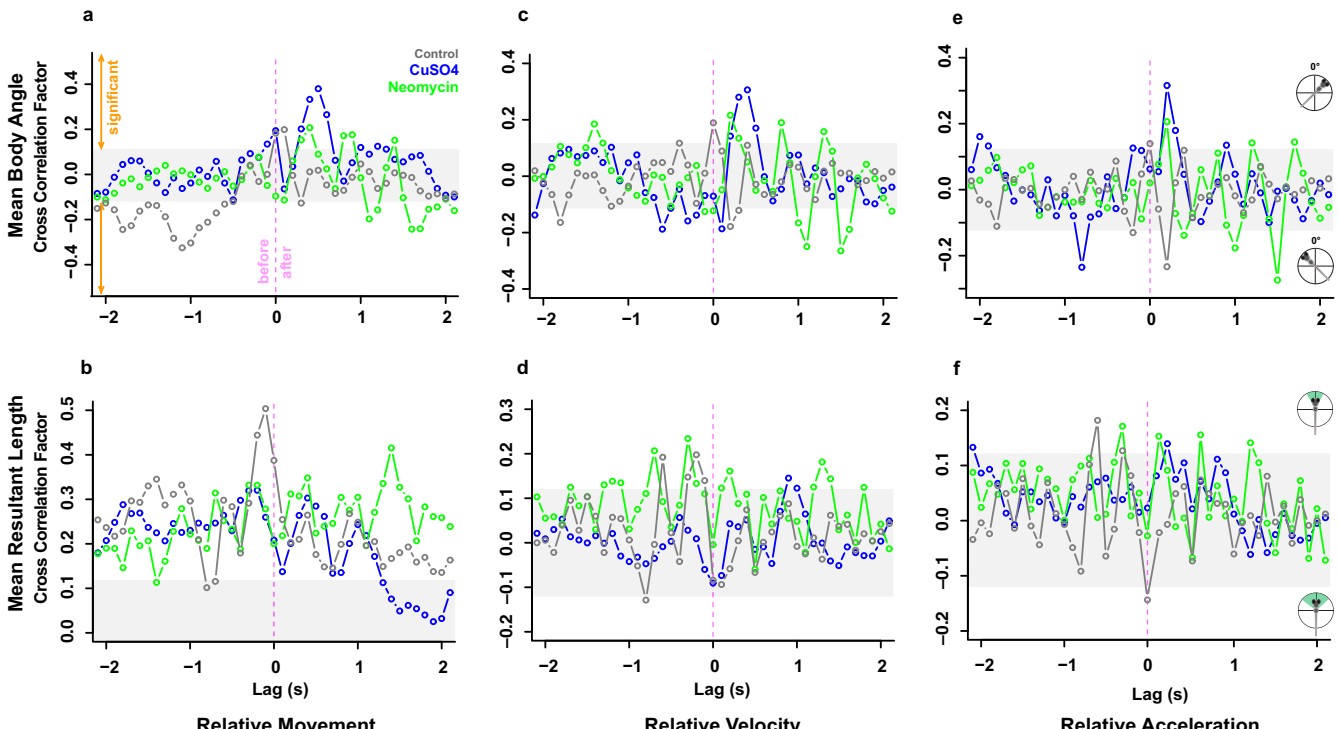

**Fig. 9 Cross correlations between linear and angular movement data indicate ototoxic compound-specific changes to the rheotaxis behavioral profile of fish.** The correlograms depict how an above average increase in relative movement (**a**, **b**), relative velocity (**c**, **d**), or relative acceleration (**e**, **f**) were significantly cross correlated with above average increases or decreases in the mean body angle (**a**, **c**, **e**) or mean resultant length (**b**, **d**, **f**). In the figures for mean body angle (**a**, **c**, **e**), the positive and negative peaks indicate that fish were oriented to the right or left of the oncoming flow vector, respectively. In the figures for mean resultant length (**b**, **d**, **f**), the positive and negative peaks indicate fish that had a lesser or greater variance of the mean body angle, respectively. The X-axis indicates the relative timing, or lag, of the cross correlation between the angular parameter with respect to an above average increase in the linear parameter (zero = simultaneous occurrence; negative = angular change occurs *before* linear change; and positive = angular change occurs *after* linear change). *For example, interpret panel 9a as:* In control (gray, $n = 248$) fish, above average increases in relative movement are most significantly correlated with changes in mean body angle to the left of the flow vector that previously occurred. In CuSO$_4$-treated (blue, $n = 204$) fish, above average increases in relative movement are most significantly correlated with above average changes in mean body angle that subsequently occurred. In neomycin-treated (green, $n = 222$) fish, above average increases in relative movement are most significantly correlated with below average changes in mean body angle that subsequently occurred.

oncoming flow vector *after* large changes in movement (Fig. 9a), velocity (Fig. 9c), and acceleration (Fig. 9e). However, neomycin-treated fish tended to change their body angle to the *left* of the flow *after* changes in movement (Fig. 9a), velocity (Fig. 9c), and acceleration (Fig. 9e). As in controls, lesioned fish always had a *reduced* angular variance correlated with changes in their linear movements, but the relative timing of the cross correlation shifted according to the ototoxic compound used and the time derivative of the linear movement. In CuSO$_4$-treated fish, the strongest correlation between a *reduction* in angular variance tended to occur *prior* to changes in movement (Fig. 9b), and *after* changes in velocity (Fig. 9d) or acceleration (Fig. 9f). Conversely, in neomycin-treated fish, *reductions* in angular variance occurred *after* changes

in movement (Fig. 9b), and *before* changes in velocity (Fig. 9d) or acceleration (Fig. 9f). These data revealed that control fish with an intact lateral line detected oncoming flow and adjusted their mean body angle *prior* to making a change in their relative movement, while lateral line lesioned fish changed their mean body angle *after* a change in their relative linear movement. Our results support the reported mechanism of water flow detection in larval zebrafish where they use their lateral line hair cells to sense the vorticity, or curl, created by gradients in the flow field that determine flow direction by measuring the temporal change in vorticity as they experience flow[22].

It appears that an intact lateral line allowed fish to rapidly detect flow and adjust their heading prior to swimming into the

flow. However, in the absence of lateral line and visual cues, and the inability of 6–7 dpf zebrafish larvae to detect horizontal angular cues (yaw) by the vestibular system[28,29], it is possible that lesioned fish might have relied on tactile cues, such as physical displacement along the substrate, to gain an external frame of reference necessary to orient and swim into flow. Intact fish also tended to change their body angle to the left of the flow vector, which may reflect lateral line handedness where larval zebrafish prefer to use the left side of their lateral line to detect flow stimuli in a manner like that of blind cavefish[39]. Lateral line ablation shifted the relative timing of the cross correlations between angular and linear movement and might have impaired any potential lateral line handedness, which may have further contributed to the erratic rheotaxis behavior observed in lesioned fish.

In summary, lateral line ablation of larval zebrafish resulted in distinct, treatment-specific rheotaxis profiles that differed from that of intact fish in the following ways: (1) delayed relative timing between changes in mean body angle and all linear movements, (2) changed mean body angle in response to flow, and (3) shifted timing between reductions in angular variance and changes in linear movement.

## Conclusions
In this study, ablating the lateral line of larval zebrafish with two commonly used ototoxic compounds impacted their ability to produce the fine adjustments required to station hold in response to water flow. Lateral line ablation inhibited the ability of fish to discriminate subtle distinctions in flow, resulting in more intense overcorrections and decreased economy of motion. Our data support the hypotheses that the lateral line mediates station holding behavior[2,33,38], but is not required for rheotaxis behavior[2,11,12,31,32] in larval zebrafish in non-uniform laminar flow. We posit that the physical displacement of lesioned fish along the substrate may have provided sufficient tactile cues[8,10] necessary for lateral line ablated fish to perform rheotaxis.

We propose that the greater angular variance observed in intact fish might indicate that these fish were regularly sampling the velocity gradients of the flow stimuli[22] from a variety of body angles so that they could reduce their response latency, quickly orient with respect to fluctuating flow stimuli, and maintain their overall mean body angle with greater fidelity to the flow vector than lesioned fish. During flow, intact fish had recurring fluctuations in relative movement, velocity, and mean body of lower magnitude, and fluctuations in relative movement, velocity, acceleration, and mean body angle of controls that were fewer in number yet occurred over a wider range of temporal frequencies compared to lesioned fish. Thus, the sensory cues detected by the lateral line allowed control fish to respond to water flow with less intensity and greater temporal variation in their movements, resulting in greater economy of movement.

This is the first study to demonstrate that two ototoxic compounds commonly used to ablate the lateral line impacts the behavioral profiles and mechanism of rheotaxis in zebrafish. We propose a unique functional assay for understanding the behavioral impacts of sensory hair cell ototoxicity, which may be used to supplement future studies exploring lateral line injury, protection, and recovery. Furthermore, the simplicity of the equipment and precision of the machine learning analyses used in this assay make it amenable to adaptation for detecting subtle behavioral changes in a wide variety of animal models.

## Methods
**Ethics statement**. This study was performed with the approval of the Institutional Animal Care and Use Committee of Washington University School of Medicine in St. Louis (Protocol number: 20-0158) and in accordance with NIH guidelines for use of zebrafish.

**Zebrafish**. Adult zebrafish were raised under standard conditions at 27–29 °C in the Washington University Zebrafish Facility. The wild type line AB* was used for all experiments unless otherwise stated. Embryos were raised in incubators at 28 °C in E3 media (5 mM NaCl, 0.17 mM KCl, 0.33 mM CaCl$_2$, 0.33 mM MgCl$_2$;[40]) with a 14:10 hr light:dark cycle. After 4 days post fertilization (dpf), larvae were raised in 100–200 mL E3 media in 250-mL plastic beakers and fed rotifers daily. Sex was not considered because it cannot be determined in zebrafish larvae at the developmental stage used in this study.

**Lateral line ablation**. At 6 or 7 dpf, ~15 larval zebrafish were placed into each well of a flat bottom 6-well polystyrene plate (#351146, Falcon) in 8 mL of E3 media. Treatment animals were placed into 50 μM neomycin trisulfate salt hydrate (N1876-25G, Sigma Aldrich) or 10 μM CuSO$_4$ (451657-10 G, Sigma Aldrich) solutions made in 8 mL of E3 media. The plate was placed into an incubator at 29 °C and exposed to the treatment for 30 min (neomycin) or 60 min (CuSO$_4$). The time courses and concentrations of neomycin and CuSO$_4$ used were chosen to introduce the maximum amount of lateral line hair cell loss with minimal damage to muscle fibers[41] and reduce non-specific toxicity[23,25], respectively. After exposure, the fish were removed from treatment, rinsed 3X in media, placed into 8 mL of clean media and allowed to recover in the incubator for 120 min (CuSO$_4$) or 150 min (neomycin) to standardize the total experimental time to 180 min. Control fish received no chemical treatments yet underwent the same procedures as the CuSO$_4$ and neomycin fish. At the end of treatment, the larvae were removed from the incubator and immediately began rheotaxis behavior trials.

**Immunohistochemistry**. Lateral line ablation was confirmed via immunohistochemistry on a subset of control and lesioned fish used in the behavior experiments. Larvae were sedated on ice (5 min, 0 °C) then fixed overnight at 4 °C in PO$_4$ buffer with 4% paraformaldehyde, 4% sucrose, and 0.2 mM CaCl$_2$. Larvae were rinsed 3X with PBS and blocked for 2 hr at room temperature in PBS buffer with 5% horse serum, 1% Triton-X, and 1% DMSO. Primary antibodies for Otoferlin (HCS-1, mouse IgG2a, 1:500, Developmental Studies Hybridoma Bank,) and Calbindin (mouse IgG1, 1:1000, Cat#:214011, Synaptic Systems) were diluted in 1x PBS buffer with 2% horse serum and 0.1% Triton-X, then incubated with the larvae overnight at 4 °C with rotation. Larvae were rinsed 5X with PBS solution, then placed in secondary antibody (goat anti-mouse IgG2a, Alexa 488, 1:1000, ThermoFisher; goat anti-mouse IgG1, Alexa 647, 1:1000, ThermoFisher) diluted in PBS with 2% horse serum and incubated for 2 hr at 22 °C with rotation. Fish were rinsed 3X with PBS then incubated with DAPI (1:2000, Invitrogen) in PBS for 20 min at 22 °C to label cell nuclei. Larvae were rinsed 2X with PBS then mounted onto glass slides with elvanol (13% w/v polyvinyl alcohol, 33% w/v glycerol, 1% w/v DABCO (1,4 diazobicylo[2,2,2] octane) in 0.2 M Tris, pH 8.5) and #1.5 cover slips.

**Confocal imaging and hair cell quantification**. Immunolabeled z-stack images were acquired via an ORCA-Flash 4.0 V3 camera (Hamamatsu) using a Leica DM6 Fixed Stage microscope with an X-Light V2TP spinning disc confocal (60 μm pinholes) controlled by Metamorph software. The region of interest tool in Metamorph was used to select specific neuromasts (~700 × 700 px) from the surrounding area. Z-stack images of 100 ms exposure were acquired through a 63X/1.4 N.A. oil immersion lens, 0.5 μm z-step size. Excitation for DAPI (405 nm) Alexa 488, and Alexa 647 was provided by 89 North LDI-7 Laser Diode Illuminator on the lowest power setting (20%) that could acquire images and minimize photobleaching. Confocal images were processed in FIJI (ImageJ, NIH) software to create maximal intensity z-stack projections with minor exposure and contrast adjustments. Hair cells were quantified by scrolling through z-stacks and counting only cells that had both DAPI and HCS-1 labels present. Any hair cells that had pyknotic nuclei (indicated by condensed DAPI labeling) were not included in the counts. For each fish, the mean number of hair cells per neuromast for the midline was calculated among multiple locations (L3-5), and for the supra orbital (SO1, SO2) neuromasts the mean was calculated between the left and right locations to account for interindividual variation. Hair cell count distributions were tested for normality; statistical significance was determined using the Kruskal-Wallis test with Dunn's multiple comparisons post hoc tests using GraphPad Prism 9.2.0.

**Experimental apparatus for rheotaxis behavior**. A microflume (220 × 100 × 40 mm; Fig. 1a) was constructed in two pieces of translucent clear resin (RS-F2-GPCL-04, Formlabs) using a high-resolution 3D printer (Form 2, Formlabs) and joined with two-stage epoxy. Flow collimators (~40 mm long) made of ~3 mm diameter plastic straws were placed immediately upstream of the working section in a portion of the flume bounded on either side by plastic mesh (25 μm) that was cemented to the wall with epoxy (Fig. 1a). Silicone sealant was used to cement a 1 mm-thick layer of plastic on top of flume to prevent water spillage from the high-pressure side of the flume by covering the pump outflow, first bend, and flow collimators up to the working section. The low-pressure portion of the flume downstream from the working section was open at the top to

facilitate the addition and removal of larval zebrafish and water when necessary. Larvae were isolated within the upper 10 mm of the flume working section to enable reliable video recording by a square (30 ×30 x 10 mm) arena that was 3D printed from clear resin. Plastic mesh (25 μm) was cemented with epoxy to the upstream and downstream sides of the removable arena which allowed water to flow through the working section after the arena was friction seated into the channel at the top of the flume and immediately adjacent to the flow collimators. An Arduino (UNO R3, Osepp) with a digital display used custom scripts to control the onset and offset of the pump and camera to ensure the consistent duration of water flow and video recordings. Flow of constant velocity was provided by a 6 V bow thruster motor (#108-01, Raboesch) inserted into the flume (Fig. 1b) and modulated by an inline rheostat between the Aurdino and pump. Methylene blue tests indicated that the flow field was laminar, non-uniform, and stable after < 250 ms of pump onset (Supplementary Fig. 3). The flow field had a lateral gradient along the Y-axis that was created by a minute boundary layer of null flow immediately adjacent to the walls and gradually increased (over ~2–5 mm, front to rear) until it became part of the freestream flow field (i.e., blue arrows in Supplementary Fig. 3).

The flume was placed onto a diffused array of 196 LEDs that emitted infrared (IR, 850 nm) light up through a layer of diffusion material (several Kimwipes© sealed in plastic) and the translucent flume (Fig. 1b). A monochromatic high-speed camera (SC1 without IR filter, Edgertronic.com) with a 60 mm manual focus macro lens (Nikon) was placed on a tripod directly over the flume to record behavioral trials at $f16$, 1/1000 s, ISO 20000, and 60 or 200 frames s$^{-1}$ (fps). Videos were saved onto 64 GB SD cards and archived on a 12 TB RAID array (see below) All source files, scripts, details of apparatus construction, and SOP provided online in the Open Source Framework repository, https://osf.io/rvyfz/.).

**Lateral line isolation and rheotaxis trials**. To isolate the contribution of the lateral line to rheotaxis, we conducted the behavioral trials under infrared light to eliminate visual cues (Fig. 1a) and reduced linear acceleration cues to the vestibular system by using a flow stimulus that rapidly accelerated (<250 ms) to a constant maximum velocity (9.74 mm s$^{-1}$). Angular acceleration was not a factor because larval zebrafish at 6–7 dpf cannot detect the angular motion of yaw in the horizontal plane[28,29]. Tactile cues were not possible to selectively block in a non-invasive manner.

The flume was filled with E3 media (28 °C) and the arena placed within the flume. A thermometer was placed into the open portion of the flume to monitor heat generated by the IR lights and miniature ice packs (2 × 2 cm; −20 °C) were used to maintain a consistent temperature range of 27–29 °C. Under IR illumination, a single larval zebrafish was transferred by pipette from the six-well plate to the arena within the flume. The swimming activity of the fish was monitored for ~10 s to ensure that it exhibited the burst-and-glide behavior indicative of normal larval zebrafish swimming[42]. The Arduino was used to begin the trial by using a custom script that triggered the camera to record 10 s of baseline swimming behavior without flow then activate the pump and record 20 s of swimming behavior under flow. After 30 s had elapsed (no flow = 0–10 s; initial flow = 10–20 s; and final flow = 20–30 s), the pump and camera turned off and the larvae was removed from the arena. Cohorts of five individual fish from each group (control, neomycin, or CuSO$_4$) were tested before switching to a new group of fish. This process was repeated for up to four iterations during each experimental session ($n = 15$–20 fish per treatment per session) for a total of 18 sessions (control, $n = 248$; CuSO$_4$, $n = 204$; neomycin, $n = 222$).

**3D markerless pose estimation (DeepLabCut)**
*Equipment.* Our behavioral data acquisition and analysis computer ran on the Windows 10 operating system and was based on a Dell Precision 3630 workstation with an Intel Xeon E-2246G processor, 64 GB RAM, multiple 2TB SSD hard drives, EVGA GFORCE RTX 2080Ti video card (GPU), dual 24″ 4 K monitors, Dell Thunderbolt 3 PCIe Card, OWC Mercury Elite Pro Dock (TB3RSDK24T) - 24TB Thunderbolt 3 Dock and Dual-Drive RAID configured as 12TB RAID 1.

*Installation.* Our GPU required Tensorflow 1.12 with the NVIDIA CUDA package to be installed prior to multi-animal DeepLabCut2.2b8 (maDLC[43,44]), Python 3.6, and necessary dependencies (https://github.com/DeepLabCut/DeepLabCut/blob/master/docs/installation.md). Detailed tutorials for using maDLC with a single animal are available online (https://github.com/DeepLabCut/DeepLabCut/blob/master/docs/maDLC_AdvUserGuide.md); however, the pertinent details of our procedure are as follows.

*Dataset curation.* To reduce computational load, all video files were downsampled to 1000 x 1000 px, dead pixels (i.e., black spots) and extraneous portions of the video were cropped out using the video editor function.

*Project creation.* A single animal maDLC project was created and the project config.yaml file was modified to create and draw a skeleton interconnecting seven unique body parts (left and right eyes, swim bladder, four points along the tail; Fig. 1b) on each larva. A curated set of ten videos provided representative examples of target positive rheotaxis behaviors across experimental treatments, 20 frames

were extracted from each video, and the seven body parts labeled on each frame. The annotated frames were checked for accuracy and multiple additional skeletal connections between the seven body parts were added to increase maDLC learning speed and model accuracy.

*Pose estimation.* The training dataset used cropped images (400 × 400 px) to reduce computational loading, Resnet-50 pre-trained network weights, and imgaug data augmentation. We trained the network until all the loss parameters plateaued at 100,000 iterations then it was evaluated (PCK values close to 1, RMSE values low) and cross-validated using the default parameters.

*Identity tracking.* The curated videos were analyzed, and the detections assembled into tracklets using the box method because it provided the best results for our test subjects. The original videos were overlaid with the newly labeled body parts to correct outliers and misidentification of body parts in the tracklet data files.

*Post processing.* The results were plotted for each video and a new labeled video was created to double check for labeling accuracy and ensure there was no need to augment the data set with additional labeled frames. Novel videos were batch processed and analyzed up through the plot trajectories and label videos step.

**Supervised behavioral annotation, classification, and analysis (SimBA)**
*Definition of rheotaxis behavior.* We created a custom Python feature extraction script (file online) that defined positive rheotaxis as when the larvae swam into the oncoming water flow at an angle of 0° ± 45° for at least 100 ms (Supplementary Fig. 1). The tail movement and forward body translation components were used to distinguish active swimming behavior into the flow from passive backward drift with a body angle of 0° ± 45°.

*Installation.* SimBAxTF-development version 68[45], Python 3.6, Git, FFmpeg, and all dependencies. (https://github.com/sgoldenlab/simba/blob/master/docs/installation.md).

*Dataset curation.* The pre-processed video files used for maDLC analysis were converted to AVI format using the SimBA video editor function.

*Project creation.* A new SimBA project was created according to the Scenario 1 tutorial: (https://github.com/sgoldenlab/simba/blob/master/docs/Scenario1.md). The user defined pose configuration and DLC-multi animal options were selected prior to generating the project config file. The user-defined pose configuration nomenclature was modified to match the body part and individual animal labels used in the maDLC config.yaml file. A curated set of ten representative rheotaxis behavior videos from each treatment group, along with their associated final tracklet data files created by maDLC, were imported and their frames extracted.

*Load project.* The project was loaded and the video parameters (frame rate (fps), resolution, and pixel measurements (px/mm)) set for each curated video. Outlier correction was achieved using the swim bladder and tail-3 body parts, movement criterion set to 0.7, and location criterion set to 1.5[45]. Features were extracted using the custom Python feature extraction script file. Rheotaxis behavior was labeled (i.e., annotations were created for predictive classifiers) for each curated video, save one that was set aside for validation. The default training, hyperparameters, and evaluation settings were used to create the model[45]. Throughout the earmarked video, the interactive plot function was used to validate the model by determining the probability threshold for the accurate prediction and minimum duration of a rheotaxis event. These values became the discrimination threshold (0.5) and minimum behavior bout length (100 ms) settings used to run the machine model. Naïve videos were processed, behaviors annotated, and machine results data files archived for analysis.

**Data analysis**. Rheotaxis only occurred when fish swam at angles 0° ± 45° under flow conditions; therefore, the rheotaxis data were compared to data when fish randomly swam at 0° ± 45° under no flow. However, the entire 0–360° body angle datasets were used to determine if the fish from each treatment group were randomly distributed under no flow (Fig. 3), and to determine the X-Y (2D) spatial use under no flow (Fig. 6a) and flow (Fig. 6b) conditions. Time series of rheotaxis data sampled at 200 and 60 Hz (fps) were quantized into their lowest common bin size of 100 ms.

**Data exclusions**. Missing data (<4%) were excluded from the analyses. In such rare cases, the tracking of individuals was not possible because the software model, despite being optimized to maximum efficacy, could not always identify the body parts of the transparent larvae under the low contrast IR illumination. These data were recorded as zeros but not included to avoid skewing the central tendency computations (e.g. means, SE).

**Statistics and reproducibility**. Sample sizes were, in total, 248 fish (control), 204 fish (CuSO4-treated), and 222 fish (neomycin-treated). Data were collected from 18 replicates. A replicate was defined as a separate experimental session. Individual animals were allocated to groups by combining all larvae into a single large beaker then randomly selecting individuals with a transfer pipette and placing them into control or treatment groups in random order.

Data wrangling and cleaning was performed in R[46] with the packages *tidyverse*[47], *dplyr*[48], *plyr*[49], and *readbulk*[50]. Figures and graphs were created with packages *circular*[51], *ggplot2*[52], and *viridis*[53]. The Rayleigh statistical tests (V-test) of uniformity for circular data in a specified mean direction ($mu = 0°$) and the Watson-Wheeler tests for differences in the grand mean body angle or angular variance (the test does not specify which parameter differs) were performed with the package *CircMLE*[54]. The mean duration, number, total distance traveled, and mean latency to the onset of rheotaxis events were calculated for 10 s bins (i.e., none, initial, and final flow) using SimBA. These data were analyzed in R using the generalized linear mixed models (GLMM) and post hoc *t*-tests (Satterthwaite method) in the packages *lmerTest*[55] and *lme4*[56]. The significance values for fixed effects of these GLMM tests are not reported by those packages and were run separately using the package *stats*[46] for type III ANOVAs. The package *zoo*[57] was used to convert rheotaxis data into times series and the package *spectral*[58] was used to perform the spectral decomposition analysis.

**Reporting summary**. Further information on research design is available in the Nature Portfolio Reporting Summary linked to this article.

## Data availability
The datasets generated and analyzed for this study are available in the Open Science Framework repository, https://osf.io/rvyfz/.

## Code availability
The R code generated for the analyses during this study are available in the Open Science Framework repository, https://osf.io/rvyfz/. Deep Lab Cut software package: https://github.com/DeepLabCut. SimBA: https://github.com/sgoldenlab/simba

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

## Acknowledgements

We want to thank Valentin Militchin for the design and construction of the electronic components of the experimental apparatus, David Lee for editorial support, and Mark Warchol for feedback on the manuscript. This work was supported by the National Institute on Deafness and Other Communication Disorders R01DC016066 (L.S.), NIDA R00DA045662 (S.A.G.), NIDA P30 DA048736 (S.R.O.N. and S.A.G.), and NARSAD Young Investigator Award 27082 (S.A.G.).

## Author contributions

Conceptualization - K.C.N. and L.S.; Methodology - K.C.N., D.K., S.R.O.N., S.A.G., and L.S.; Data acquisition - K.C.N. and A.L.S.; Data analysis - K.C.N., D.K., S.R.O.N., and A.L.S.; Writing (original draft) - K.C.N; Writing (revisions) - K.C.N. and L.S. with feedback from S.R.O.N. and S.A.G.; Project supervision - L.S.; Funding acquisition - S.R.O.N., S.A.G., and L.S.

## Competing interests

The authors declare no competing interests.
