## [Peer Review File · Communications Biology]

Reviewers' comments:

Reviewer #1 (Remarks to the Author):

The manuscript by Newton et al. studies the changes in zebrafish larvae behavior induced by exposure to chemicals that are commonly used for lateral line ablation. It essentially reports that fish still perform rheotaxis after chemical ablation but in a clumsier fashion, and that the two chemicals have different effects on the resulting behavior.

I have severe concerns over the methods employed to demonstrate the results claimed in the manuscript. Given that ablation of the lateral line is at the core of all the interpretation and discussion of this manuscript, one would expect that it is extensively checked and quantified. However, there is only one figure (supp fig 1), showing only one sample of damaged neuromast per chemical. This is clearly insufficient to allow the authors to build their conclusions beyond reasonable doubt.

For instance, figure 3 shows that the larvae still perform some kind of rheotaxis after treatment, and without visual cues. Before going further and hypothesize that tactile cues are responsible for this behavior, I would have expected:

1) Rock-solid proof (including quantification) that the neuromasts are actually completely or partially destroyed over a significant portion of the tested individuals. This could be done for instance by generalizing the immunohistochemical treatment presented in the manuscript, or with a simpler protocol with DASPEI, and image processing of the result under confocal imaging.

2) Rock-solid proof (including quantification) that the larvae actually use tactile cues when the neuromasts are ablated. This could be done in a rather simple manner by imaging the fish from the side, as reported in several studies cited by the authors, and then counting the contacts with floor and ceiling and/or establishing causality between contacts and forward motion.

Then, the authors see a behavioral difference after exposure to CuSO₄ and Neomycin. A partial ablation of the LL by Neomycin is hypothesized but not checked, despite being a crucial point for the study. Personal experimental observations of the impact of Neomycin over neuromasts have led me to conclude that different batches of neomycin bought at different times and stored in different conditions can lead to very different results. I am therefore not surprised to see that its effect is milder than CuSO₄ in this study, but I also suspect that different results would be obtained with different batches, even from the same supplier. It is not stated in the methods how many batches of Neomycin have been used, but the authors should try different batches or, if this is already the case, group their data accordingly and quantify the variability. In the present state, I cannot reasonably consider that the behavioral differences reported in this paper are indeed due to a difference between CuSO₄ and Neomycin and not to a deficient batch of Neomycin.

In addition, should the authors decide to resubmit their manuscript including the aforementioned checks, I have several other comments that would require major modifications. Here is a shortlist of the most important ones:

- Many details on the experimental setup are missing, and the scheme of the setup is too sketchy to understand the experimental details. What is the 3D shape of the working section? How is it sealed to/seating onto the channel? How thick are the filters? Where is the rheostat positioned? How stable is the flow in the working chamber?

- The statistics over the number of individuals per experiment are not clear. The methods specify four cohorts of five individuals, which makes 20 fish per group. This is certainly too few to draw any conclusion, and in contradiction with the figures, e.g. fig.2 seems to contain approx. 250 points per condition. Much more details should be provided in the methods, and the number of fish used for each plot should be stated in the figure legends.

- As shown in figures 5 and 6, the larvae tend to stay close to the walls. This is clearly an issue since the flow velocity is always null at the walls. The length of the boundary layer should be

determined (either analytically or by measurements) and moments where the fish are too close to the wall (i.e. subject to significantly reduced flow or high shear) should be discarded from the analysis.

- Many details are provided on the usage of DeepLabCut and Simlab in the methods, but unfortunately the most important information is lacking (or so hidden/unclearly formulated I couldn't find it): how are the training sets defined, and how large are they?

I therefore cannot recommend publication of this manuscript in the present state, and ask for both convincing evidence over the main methodological flaws cited above and revision of the manuscript before going further in the reviewing.

Other comments:

- CuSO₄ is not per se a toxin since it is not produced by a living organism. Please update the wording.

- Using DeepLabCut and SimLab for performing these analysis is completely overkill. This is not crippling for publication, but from a reader's perspective it makes the methods extremely long and difficult to read, for no reason.

- Figure 2 is almost useless, it should be merged with figure 1 or moved to supplementary.

- Figure 5 and 6 are highly redundant, one should be moved to supplementary.

Reviewer #2 (Remarks to the Author):

In my view, the ms meets the criteria for publication in Communications Biology. It provides a significant advance with novel experimental results, in depth data analysis, and innovative experimental and computational methods. The ms contains appropriate reference to preceding publication in this area, and an appropriately nuanced coverage of this complex multisensory behavior.

I didn't find any technical or conceptual flaws. The ms is of relevance to the community using Zebrafish a model system for the study of haircell function but also to the wider community of fish sensory neuroethology.

My only criticism of the ms is that it should be clear that these results are specific to Zebrafish and the circumstances of testing. This should be explicit in the title of the paper – and in other cases in the ms where the term 'fish' is used but where it should be more specifically 'zebrafish'. There is a huge diversity of fish species (and developmental stages), where both the drug and behavioral effects will differ. Behavioral effects will also depend on such things as the test flume characteristics and use single or multiple individuals in testing. I agree with the authors' conclusion that their "novel functional assay for hair cell ototoxicity,may be used to supplement future studies exploring lateral line injury, protection, and recovery." But some of the detailed analysis is of less relevance to the sensory neuroethology community and some could perhaps be moved to the supplementary data section to broaden the appeal of the main body of the ms to a wider audience.

Lateral Line Ablation by Ototoxic Compounds Results in Distinct Rheotaxis Profiles in Larval Zebrafish

Kyle C Newton^{1,5*}, Dovi Kacev², Simon R O Nilsson³, Allison L. Saettele¹, Sam A Golden³, and Lavinia Sheets^{1,4*}

Author Response to Reviewers Comments:

We want to express our sincere gratitude to the reviewers who took time out of their busy schedules to make constructive comments and help improve our manuscript. We endeavored to address each concern and accommodate all suggestions except in the rare occasion when a suggestion was not possible or did not improve the overall manuscript. The following is a point-by-point reply to all the suggestions from each reviewer along with our corrective actions and rationale for our responses.

Referee expertise:

Referee #1: zebrafish, behaviour, rheotaxis

Referee #2: marine biology, rheotaxis

Reviewers' comments:

Reviewer #1 (Remarks to the Author):

The manuscript by Newton et al. studies the changes in zebrafish larvae behavior induced by exposure to chemicals that are commonly used for lateral line ablation. It essentially reports that fish still perform rheotaxis after chemical ablation but in a clumsier fashion, and that the two chemicals have different effects on the resulting behavior.

I have severe concerns over the methods employed to demonstrate the results claimed in the manuscript. Given that ablation of the lateral line is at the core of all the interpretation and discussion of this manuscript, one would expect that it is extensively checked and quantified. However, there is only one figure (supp fig 1), showing only one sample of damaged neuromast per chemical. This is clearly insufficient to allow the authors to build their conclusions beyond reasonable doubt.

For instance, figure 3 shows that the larvae still perform some kind of rheotaxis after treatment, and without visual cues. Before going further and hypothesize that tactile cues are responsible for this behavior, I would have expected:

1) Rock-solid proof (including quantification) that the neuromasts are actually completely or partially destroyed over a significant portion of the tested individuals. This could be done for instance by generalizing the immunohistochemical treatment presented in the manuscript, or with a simpler protocol with DASPEI, and image processing of the result under confocal imaging.

Author Response:

In our previous manuscript, lesion of neuromasts in tested individuals was qualitatively checked, but not quantified. In this revised manuscript, we provide statistically significant evidence of complete (CuSO₄) and partial (nearly complete; neomycin) neuromast ablation in larvae that were tested for rheotaxis behavior (revised Figure 2). Individuals from three separate behavioral

trials (4-6 fish per condition per trial) were previously fixed, immunolabeled, and visually examined to confirm neuromast lesion (representative examples shown in previous Supp. Figure 1). We reevaluated these same specimens by acquiring high resolution confocal images of lateral-line neuromasts (SO1 and SO2, and posterior L3-L5) and quantifying the number of hair cells per neuromast per individual fish. These results provide proof that a portion of tested larvae showed complete neuromast hair cell loss with CuSO₄ treatment and severe hair cell loss following neomycin treatment. We have added the following text and revised figure to the manuscript:

Line 88: We observed near total hair cell loss in both anterior and posterior lateral line neuromasts with CuSO₄ treatment (Fig. 2 a, b, d, e, g) and a significant impact on the morphology and reduction in hair cell number per neuromast following neomycin treatment (Fig. 2 a, c, d, f, g). These results support that the ototoxic compound-treated fish used in our behavior assay had a total absence or severe impairment of lateral line function.

Figure 2. Confirmation of neuromast hair cell loss following CuSO₄ or neomycin treatment. a-f) Representative confocal max intensity projection images of the: a-c) mid posterior lateral line (MidLL) fourth neuromast (L4); and d-f) second anterior supraorbital (SO2) neuromast from the fish cohorts used for behavior experiments. Hair cells were labeled with an antibody against Otoferlin (HCS1; green a-f, gray a'-f'). Afferent neurons were labeled with an antibody against Calbindin (magenta), and cell nuclei were labeled with DAPI (blue). g) Quantification of the grand mean (\pm SEM) number of hair cells per neuromast in intact (CTL), CuSO₄- and neomycin-treated fish. Each dot represents the mean number of hair cells from the MidLL (L3, L4, and L5) or SO (left and right) neuromasts from an individual fish. Data were collected from fish used in three experimental behavior trials; 4-6 fish per condition per trial. Significance values: ** < 0.01, *** < 0.001, **** < 0.0001

Reviewer #1:

2) Rock-solid proof (including quantification) that the larvae actually use tactile cues when the neuromasts are ablated. This could be done in a rather simple manner by imaging the fish from the side, as reported in several studies cited by the authors, and then counting the contacts with floor and ceiling and/or establishing causality between contacts and forward motion.

Author Response:

Given the design of our behavior apparatus, we could not test whether the fish used tactile cues. The micro flume was created with translucent (not transparent) plastic and filmed with a macro lens set at significant barrel extension (resulting in shallow depth of field despite small aperture) to fill the frame of the image with the arena/working section, all of which make it

impossible to clearly view fish from the side. We therefore amended all references to tactile cues to be speculative i.e., to read that “tactile cues might have been used but we could not test this hypothesis.”

Line 131: However, we did not eliminate tactile cues because we could not prevent fish from contacting the substrate and our flow rate was sufficient to displace substrate coupled fish along the bottom and against the rear mesh of arena. Therefore, we propose that lateral line ablated fish might have used tactile cues to gain an external frame of reference and perform rheotaxis (8, 10), but explicitly testing this idea would require modifying our experimental apparatus to enable the video capture of fish movements along the vertical plane (Z-axis).

Line 271: The propensity of lesioned fish to use the rear of the arena while performing rheotaxis in the absence of visual (3) and horizontal vestibular (31, 32) cues suggests that these larvae might have used tactile cues to provide the external frame of reference necessary to orient and swim against flow (2, 8, 10). Unfortunately, our flume design and camera setup prevented us from exploring this possibility.

Line 505: It appears that an intact lateral line allowed fish to rapidly detect flow and adjust their heading prior to swimming into the flow. However, in the absence of lateral line and visual cues, and the inability of 6-7dpf zebrafish larvae to detect horizontal angular cues (yaw) by the vestibular system (^{31, 32}), lesioned fish might have relied on tactile cues, such as physical displacement along the substrate, to gain an external frame of reference necessary to orient and swim into flow.

Reviewer #1:

Then, the authors see a behavioral difference after exposure to CuSO4 and Neomycin. A partial ablation of the LL by Neomycin is hypothesized but not checked, despite being a crucial point for the study....

Author Response:

We have quantified the degree of ablation in a subset of behavior tested fish and observe a significant reduction, but not total absence, of neuromast hair cells following neomycin treatment. These results are shown in the revised Figure 2.

Reviewer #1:

Personal experimental observations of the impact of Neomycin over neuromasts have led me to conclude that different batches of neomycin bought at different times and stored in different conditions can lead to very different results. I am therefore not surprised to see that its effect is milder than CuSO4 in this study, but I also suspect that different results would be obtained with different batches, even from the same supplier. It is not stated in the methods how many batches of Neomycin have been used, but the authors should try different batches or, if this is already the case, group their data accordingly and quantify the variability. In the present state, I cannot reasonably consider that the behavioral differences reported in this paper are indeed due to a difference between CuSO4 and Neomycin and not to a deficient batch of Neomycin.

Author Response:

We used two different batches of neomycin during the course our 18 separate experimental sessions and observed no differences in the ability to perform rheotaxis in larval zebrafish.

Regarding the reviewer's personal experimental observations: our lab has used many different batches of neomycin for numerous experiments and has not observed notable batch-to-batch

variation in lateral line neuromast ablation that the reviewer experienced. What we have observed is an appreciable influence of different external buffers/ fish water/ embryo media on the degree of hair cell loss. In this study, we exclusively used E3 (recipe provided in the methods section) for both lateral line ablation and behavior experiments.

Reviewer #1:

In addition, should the authors decide to resubmit their manuscript including the aforementioned checks, I have several other comments that would require major modifications. Here is a shortlist of the most important ones:

- Many details on the experimental setup are missing, and the scheme of the setup is too sketchy to understand the experimental details. What is the 3D shape of the working section? How is it sealed to/seating onto the channel? How thick are the filters? Where is the rheostat positioned? How stable is the flow in the working chamber?

Author Response:

We provided more detail on the apparatus.

Line 615: A microflume (220 x 100 x 40 mm; Fig. 1a) was constructed in two pieces of translucent clear resin (RS-F2-GPCL-04, Formlabs) using a high-resolution 3D printer (Form 2, Formlabs) and joined with two-stage epoxy. Flow collimators (~40mm long) made of ~3mm diameter plastic straws were placed immediately upstream of the working section in a portion of the flume bounded on either side by plastic mesh (25 μ m) that was cemented to the wall with epoxy (Fig. 1a). Silicone sealant was used to cement a 1 mm-thick layer of plastic on top of flume to prevent water spillage from the high-pressure side of the flume by covering the pump outflow, first bend, and flow collimators up to the working section. The low-pressure portion of the flume downstream from the working section was open at the top to facilitate the addition and removal of larval zebrafish and water when necessary. Larvae were isolated within the upper 10 mm of the flume working section to enable reliable video recording by a square (30 x 30 x 10 mm) arena that was 3D printed from clear resin. Plastic mesh (25 μ m) was cemented with epoxy to the upstream and downstream sides of the removable arena which allowed water to flow through the working section after the arena was friction seated into the channel at the top of the flume and immediately adjacent to the flow collimators. An Arduino (UNO R3, Osepp) with a digital display used custom scripts to control the onset and offset of the pump and camera to ensure the consistent duration of water flow and video recordings. Flow of constant velocity was provided by a 6V bow thruster motor (#108-01, Raboesch) inserted into the flume (Fig. 1b) and modulated by an inline rheostat between the Arduino and pump. Methylene blue tests indicated that the non-uniform laminar flow field was stable after < 250 ms of pump onset (data not shown; Supplementary Fig. 3)

Reviewer #1:

- The statistics over the number of individuals per experiment are not clear.

Author Response:

We have clarified the statistics section to emphasize which R packages and which tests were used for each analysis.

Line 746: Data wrangling and cleaning was performed in R (46) with the packages *tidyverse* (47), *dplyr* (48), *plyr* (49), and *readbulk* (50). Figures and graphs were created with packages *circular* (51), *ggplot2* (52), and *viridis* (53). The Rayleigh statistical tests (V-test) of uniformity for circular data in a specified mean direction ($\mu = 0^\circ$) and the Watson-Wheeler tests for

differences in the grand mean body angle or angular variance (the test does not specify which parameter differs) were performed with the package *CircMLE* (54). The mean duration, number, total distance travelled and mean latency to the onset of rheotaxis events were calculated for 10 s bins (i.e., none, initial, and final flow) using SimBA. These data were analyzed in R using the generalized linear mixed models (GLMM) and post hoc t-tests (Satterthwaite method) in the packages *lmerTest* (55) and *lme4* (56). The significance values for fixed effects of these GLMM tests are not reported by those packages and were run separately using the package *stats* (46) for type III ANOVAs. The package *zoo* (57) was used to convert rheotaxis data into times series and the package *spectral* (58) was used to perform the spectral decomposition analysis.

Reviewer #1:

The methods specify four cohorts of five individuals, which makes 20 fish per group. This is certainly too few to draw any conclusion, and in contradiction with the figures, e.g. fig.2 seems to contain approx. 250 points per condition. Much more details should be provided in the methods, and the number of fish used for each plot should be stated in the figure legends.

Author Response:

There were 18 experimental sessions containing 5 fish per group with up to 4 iterations per session. We have clarified this information in the revised methods. We have also indicated the sample size per group and number of sessions in the methods and in the legends of Figs 3-10:

Line 665: Cohorts of five individual fish from each group (control, neomycin, or CuSO₄) were tested before switching to a new group of fish. This process was repeated for up to four iterations during each experimental session (n = up to 20 fish per treatment per session) for a total of 18 sessions (control, n = 248; CuSO₄, n = 204; neomycin, n = 222).

Reviewer #1:

- As shown in figures 5 and 6, the larvae tend to stay close to the walls. This is clearly an issue since the flow velocity is always null at the walls. The length of the boundary layer should be determined (either analytically or by measurements) and moments where the fish are too close to the wall (i.e. subject to significantly reduced flow or high shear) should be discarded from the analysis.

Author Response:

We have added additional details to the manuscript that address this concern. Specifically, we describe how we determined the length of the boundary layer and elaborate on how SimBA excludes fish that do not move their tail or swim forward when identifying positive rheotaxis behavior.

Flow field gradient and rheotaxis criteria

Line 247: The reduced velocity of the laminar flow gradient along the sides was created by a minute boundary layer of null flow adjacent to the wall that gradually increased (over ~2-5 mm, front to rear) until it became part of the freestream flow field (i.e., blue arrows in Supplementary Fig. 3). The gradient provided a refuge (e.g., 37) where intact and lesioned fish could swim into the flow with reduced energetic cost (38). As our definition of rheotaxis (see methods; Supplementary Fig. 1d, e) specifies that only fish that moved their tail and had forward body

translation into the flow, fish that were in null water flow (i.e., too close to the wall) were excluded from the rheotaxis dataset.

Boundary layers

Line 633: Methylene blue tests indicated that the flow field was laminar, non-uniform, and stable after < 250 ms of pump onset (data not shown; Supplementary Fig. 3). The flow field had a lateral gradient along the Y-axis that was created by a minute boundary layer of null flow immediately adjacent to the walls and gradually increased (over ~2-5 mm, front to rear) until it became part of the freestream flow field (i.e., blue arrows in Supplementary Fig. 3).

Rheotaxis definition automatically discards fish that stop swimming regardless of body angle:

Line 709: *Definition of Rheotaxis Behavior:* We created a custom Python feature extraction script (file online) that defined positive rheotaxis as when the larvae swam into the oncoming water flow at an angle of $0^\circ \pm 45^\circ$ for at least 100 ms (Supplemental Fig. 1). The tail movement and forward body translation components were used to distinguish active swimming behavior into the flow from passive backward drift with a body angle of $0^\circ \pm 45^\circ$.

Figure Legend: Supplementary Fig 1 (formerly Fig 2): Definition of positive rheotaxis behavior. a) Larval zebrafish in the microflume arena performing rheotaxis under flow as defined by multiple conditions, including b) the water flow stimulus was on; c) the fish body angle was oriented to $0^\circ \pm 45^\circ$ (green shaded wedge); d) the tail moved laterally every 100 ms; and e) the body of the fish had forward translation every 100 ms. Note that conditions (d) and (e) were used to discriminate between fish displaying positive rheotaxis and those passively drifting backward with body angles of $0^\circ \pm 45^\circ$.

This figure legend details the criteria used to define rheotaxis in the SimBA custom feature extraction script for behavioral annotation. Furthermore, it illustrates that regardless of their location in the arena (i.e., along the wall or in the center) fish who do not move their tail or swim forward will not be counted in the rheotaxis data even if they are oriented to $0 \pm 45^\circ$.

Reviewer #1:

- Many details are provided on the usage of DeepLabCut and Simlab in the methods, but unfortunately the most important information is lacking (or so hidden/unclearly formulated I couldn't find it): how are the training sets defined, and how large are they?

Author Response:

The training dataset is described in the methods:

Line 686: *Project Creation:* A single animal maDLC project was created and the project config.yaml file was modified to create and draw a skeleton interconnecting seven unique body parts (left and right eyes, swim bladder, four points along the tail; Fig. 1a) on each larva. A curated set of ten videos provided representative examples of target positive rheotaxis behaviors across experimental treatments, 20 frames were extracted from each video, and the seven body parts labeled on each frame. The annotated frames were checked for accuracy and multiple additional skeletal connections between the seven body parts were added to increase maDLC learning speed and model accuracy.

Pose Estimation: The training dataset used cropped images (400 x 400 px) to reduce computational loading, Resnet-50 pre-trained network weights, and imgaug data augmentation. We trained the network until all the loss parameters plateaued at 100,000 iterations then it was evaluated (PCK values close to 1, RMSE values low) and cross-validated using the default parameters.

As the reviewer previously mentioned, the methods section could easily be bogged down in machine learning procedural minutiae, so we also referred the reader to online tutorials for finer details:

Line 680: Detailed tutorials for using maDLC with a single animal are available online (https://github.com/DeepLabCut/DeepLabCut/blob/master/docs/maDLC_AdvUserGuide.md);

Reviewer #1:

Other comments:

- CuSO₄ is not per se a toxin since it is not produced by a living organism. Please update the wording.

Author Response:

“Toxin” has been removed and “ototoxic compound” used throughout the manuscript.

Reviewer #1:

- Using DeepLabCut and SimLab for performing these analysis is completely overkill. This is not crippling for publication, but from a reader's perspective it makes the methods extremely long and difficult to read, for no reason.

Author Response:

We disagree. DLC and SimBA are the reason why we could parse out the subtle yet consistent compound-specific differences in the behavioral profiles of among lesioned fish groups that cannot be reliably quantified using traditional human-based scored methods.

The NIH strongly emphasizes that researchers develop more reproducible behavior studies. A recent paper:

<https://www.sciencedirect.com/science/article/pii/S0003347209001730?via%3Dihub>

highlights how these machine learning approaches will be critical for future experiments using more advanced recording and manipulation approaches that require more temporal precision than human annotation.

The reviewer suggests that our use of open-source machine learning tools for behavioral analysis is redundant, increases complexity, and may obscure the main messages of the manuscript. We agree with the reviewer that the method section was unnecessarily lengthy, and we have streamlined the methods describing our DeepLabCut and SimBA machine learning analysis pipeline to roughly ~350 words per section for brevity. Additional detailed SOPs are located online in the Open Science Framework repository.

An alternative computational approach, that allows for some, albeit not all, of the analyses within the manuscript, is the use of prevalent commercial tools (e.g., Ethovision). However, this alternative is costly (~\$10,000 USD) and inaccessible for all but the most well-funded labs.

A further alternative would be to manually analyze and annotate all the videos using carefully set behavioral operational definitions for 674 videos (~337 hrs). However, this does not allow for immediate, detailed, behavioral analysis involving frequency decompositions, angles, etc. It would also be extremely arduous, time-consuming and non-replicable.

Importantly, through our use of open-source machine learning tools, we can share our classifier, annotations and larger analysis with the wider scientific community. This ensures that our methods are fully transparent and replicable, and that future studies interested in similar behaviors can build on and exploit our models and data. Future studies, working in similar environments, can spend less or no time manually analyzing rheotaxis behavior or creating pose-estimation models, which we believe is a significant contribution to the field.

Reviewer #1:

- *Figure 2 is almost useless, it should be merged with figure 1 or moved to supplementary.*

Author Response:

We disagree that this figure is useless because it graphically details the criteria used to define rheotaxis in the SimBA custom feature extraction script for behavioral annotation. Furthermore, it illustrates that regardless of their location in the arena (i.e., along the wall or in the center) fish who do not move their tail or swim forward will not be counted in the rheotaxis data even if they are oriented to $0\pm 45^\circ$.

This information helps address the reviewer's previous comment "*moments where the fish are too close to the wall (i.e. subject to significantly reduced flow or high shear) should be discarded from the analysis.*"

These points warrant that the figure remain in the manuscript, but we have placed it in the supplemental section.

Reviewer #1 comments:

- *Figure 5 and 6 are highly redundant, one should be moved to supplementary.*

Author Response:

Figure 5 was moved to Supplementary

Reviewer #2 (Remarks to the Author):

In my view, the ms meets the criteria for publication in Communications Biology. It provides a significant advance with novel experimental results, in depth data analysis, and innovative experimental and computational methods. The ms contains appropriate reference to preceding publication in this area, and an appropriately nuanced coverage of this complex multisensory behavior.

I didn't find any technical or conceptual flaws. The ms is of relevance to the community using Zebrafish a model system for the study of haircell function but also to the wider community of fish sensory neuroethology.

Author Response:

Thank you, we genuinely appreciate your feedback and perspective.

Reviewer #2

My only criticism of the ms is that it should be clear that these results are specific to Zebrafish and the circumstances of testing. This should be explicit in the title of the paper – and in other cases in the ms where the term 'fish' is used but where it should be more specifically 'zebrafish'. There is a huge diversity of fish species (and developmental stages), where both the drug and behavioral effects will differ. Behavioral effects will also depend on such things as the test flume characteristics and use single or multiple individuals in testing.

Author Response:

We changed the title to reflect the emphasis on larval zebrafish (from fish); however, we did not feel that changing every instance of the word "fish" to "zebrafish" was warranted. We believe when we say fish that we are not necessarily expanding our results toward other species. Furthermore, because we often refer to our subjects as "neomycin-treated fish" we believe that it becomes grammatically onerous and tedious to read "neomycin-treated larval zebrafish" and so we condensed the language for readability. We have used this terminology in previous larval zebrafish publications without complaint, but if this issue is a major sticking point, then we will reconsider our position.

Reviewer #2

I agree with the authors' conclusion that their "novel functional assay for hair cell ototoxicity,may be used to supplement future studies exploring lateral line injury, protection, and recovery." But some of the detailed analysis is of less relevance to the sensory neuroethology community and some could perhaps be moved to the supplementary data section to broaden the appeal of the main body of the ms to a wider audience.

Author Response:

We believe that the detailed analysis is one of the core values brought about by using machine vision and learning techniques because we show that common analyses in rheotaxis studies do not tell a complete and nuanced story. Furthermore, the details are the basis for our ultimate conclusions that ototoxic compound ablation changes the rheotaxis behavioral profile of larval zebrafish in compound specific ways.

Considering the reviewer's comments, we have streamlined the results & conclusions sections to make the language clearer and more accessible to a broader audience.

Reviewers' comments:

Reviewer #3 (Remarks to the Author):

It is interesting to study the effects and mechanisms mediated by Ototoxic Compounds using model animal. I have some suggestions and comments for the authors' concern.

1. The authors put a lot of introduction of the methods in the Results and Discussion Section, especially Figure 1 and related description.
2. Two compounds were used to get the impaired fish. The fixed concentration and exposure period were used. I am wondering if other concentrations can be used to make the model with different degree of damage.
3. With exception of hair cell organ damage, were there any other effects induced by the two compounds? I mean how to exclude other cause-effect relation rather than through hair cell organ damage.
4. In Fig. 4, it is a little hard to understand the significant marks, especially in b and c.
5. The authors used 8 subtitles to show the results and corresponding conclusions. It sounds sparse for the readers. I suggest that the authors put some similar results under a general subtitle so as to make the readers clearly follow the main findings of these papers. Now, it is likely for the readers to see the rich results but not to grasp the key points of this paper.

Lateral Line Ablation by Ototoxic Compounds Results in Distinct Rheotaxis Profiles in Larval Zebrafish

Kyle C Newton^{1,5*}, Dovi Kacev², Simon R O Nilsson³, Allison L. Saettele¹, Sam A Golden³, and Lavinia Sheets^{1,4*}

Author Response to Reviewers Comments:

We want to express our sincere gratitude to the reviewer who evaluated our revised manuscript. The following is a point-by-point reply to their evaluation.

Reviewer #3 comments to the authors:

It is interesting to study the effects and mechanisms mediated by Ototoxic Compounds using model animal. I have some suggestions and comments for the authors' concern.

1. The authors put a lot of introduction of the methods in the Results and Discussion Section, especially Figure 1 and related description.

Our rationale for introducing and expanding on our methodological approach in both the results and the discussion sections was that our study introduces a novel, open access technique to evaluate rheotaxis behavior in larval zebrafish. This more sensitive and quantitative approach we believe resolves the specific question of how the lateral line contributes to positive rheotaxis and provides a powerful tool that will be beneficial to other researchers investigating lateral line mediated behavior.

2. Two compounds were used to get the impaired fish. The fixed concentration and exposure period were used. I am wondering if other concentrations can be used to make the model with different degree of damage.

Many previous studies, including a few from our own lab, have evaluated neuromast damage resulting from CuSO₄ and neomycin, and have established dose- and time- response curves for both compounds. The concentrations and exposure periods used for this study were chosen as the minimum needed to induce near total ablation of lateral line neuromast hair cells.

We agree that exploring how partial damage of neuromasts impacts positive rheotaxis behavior is an interesting question, and we intend to address it in future studies.

3. With exception of hair cell organ damage, were there any other effects induced by the two compounds? I mean how to exclude other cause-effect relation rather than through hair cell organ damage.

We chose the concentrations and exposure times for both compounds based on previous studies that defined the minimum doses for either neomycin or CuSO₄ needed to introduce profound lateral line hair cell loss while causing minimal off target effects. We have added the following text to the methods.

Line 587-590: "The time courses and concentrations of neomycin and CuSO₄ used were chosen to introduce the maximum amount of lateral line hair cell loss with minimal damage to muscle fibers (⁴¹) and reduce non-specific toxicity (^{23,25}), respectively."

4. In Fig. 4, it is a little to hard to understand the significane marks, especially in b and c.

We have revised Figure 4 to make the significance marks more understandable.

5. The authors used 8 subtitles to show the results and corresponding conclusions. It sounds sparse for the readers. I suggest that the authors put some similar results under a general subtitle so as to make the readers to clearly to follow the main findings of these paper. Now, it is likely for the readers to see the rich results but not to grasp the key points of this paper.

To provide clarity and assist the readers in following the main findings of our study, we have added an additional paragraph at the end of the introduction that summarizes the key points of the paper.

Line 69-79 “We found that lateral line ablation by neomycin and CuSO₄ did not eliminate rheotaxis behavior but affected the behavioral profiles of larval zebrafish in both general and compound-specific ways. While fish with an intact lateral line could efficiently and effectively maintain their position near the source of the water flow, lateral line ablated fish occupied the rear portion of the working section, performed more rheotaxis events of shorter duration and travelled greater distances. Furthermore, exposure to ototoxic compounds reduced the temporal and angular variation in the swimming kinematics such that lateral line ablated larvae performed rheotaxis with more movements of greater intensity but with reduced efficacy. These results support that sensory input from lateral line organs contribute to effective positive rheotaxis behavior by allowing fish to detect subtle changes in water flow and to respond with greater economy of movement.”

REVIEWERS' COMMENTS:

Reviewer #3 (Remarks to the Author):

The authors have revised the manuscript carefully, and the paper can be accepted.

Lateral Line Ablation by Ototoxic Compounds Results in Distinct Rheotaxis Profiles in Larval Zebrafish

Kyle C Newton^{1,5*}, Dovi Kacev², Simon R O Nilsson³, Allison L. Saettele¹, Sam A Golden³, and Lavinia Sheets^{1,4*}

Author Response to Reviewers Comments:

Reviewer #3 comments to the authors:

The authors have revised the manuscript carefully, and the paper can be accepted.

Thank you for your thoughtful evaluation and improving the quality of our manuscript.